# CLPO: CURRICULUM LEARNING MEETS POLICY OPTIMIZATION FOR LLM REASONING

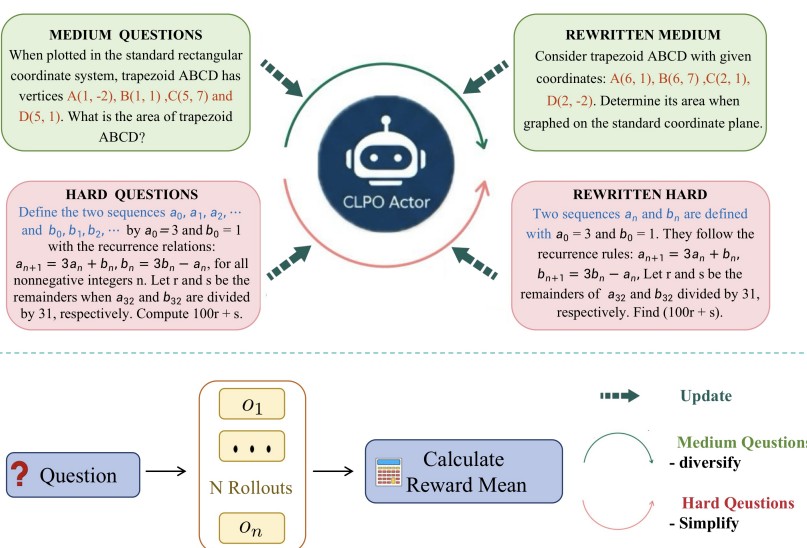

Figure 1: An overview of the Guided Self-Evolution paradigm in our proposed framework CLPO (Curriculum-guided Learning for Policy Optimization). For each initial question, CLPO first assesses its difficulty by evaluating N on-policy rollouts. The model (CLPO Actor) then acts as its own teacher, restructuring identified medium problems for diversification and hard problems for simplification. Restructured for effective learning, these new problems are then used for the final policy update, forming a dynamic, self-improving learning loop.

## ABSTRACT

Recently, online Reinforcement Learning with Verifiable Rewards (RLVR) has become a key paradigm for enhancing the reasoning capabilities of Large Language Models (LLMs). However, existing methods typically treat all training samples uniformly, overlook the vast differences in problem difficulty relative to the model's current capabilities. This uniform training strategy leads to inefficient exploration of problems the model has already mastered, while lacking effective guidance on the problems that are challenging its abilities the most, limiting both learning efficiency and the performance upper-bound. To address this, we propose **CLPO (Curriculum-guided Learning for Policy Optimization)**, a novel algorithm that creates a dynamic pedagogical feedback loop within the policy optimization process. The core of CLPO is to leverage the model's own rollout performance to conduct real-time difficulty assessment, thereby constructing an **Online Curriculum**. This curriculum then guides an **Adaptive Problem Restructuring** mechanism, where the model acts as its own teacher: it diversifies medium-difficulty problems to promote generalization and simplifies hard problems to make them more accessible. Our approach transforms the static training procedure into a dynamic process that co-evolves with the model's capabilities. Experiments show that CLPO achieves **state-of-the-art (SOTA)** performance across eight challenging mathematical and general reasoning benchmarks,

with an average **pass@1** improvement of **6.96%** over ohter methods, demonstrating its potential for more efficiently training more capable reasoning models.

# 1 INTRODUCTION

The capabilities of Large Language Models (LLMs) (Brown et al., 2020; Touvron et al., 2023; Achiam et al., 2023) have advanced substantially, demonstrating remarkable performance on a wide range of natural language tasks. However, tasks demanding complex, multi-step reasoning—such as advanced mathematical problem-solving, scientific inquiry, and code generation—continue to pose a significant challenge. While traditional Supervised Fine-Tuning (SFT) (Ouyang et al., 2022) plays a crucial role in aligning initial model capabilities, it relies on static datasets curated by experts, which is not only costly but also struggles to cover all possible reasoning paths. To overcome these bottlenecks, Reinforcement Learning (RL) (Sutton et al., 1998) has emerged as a key post-training paradigm (Ouyang et al., 2022). Unlike passively imitating static examples, RL enables models to actively learn from their own generations through exploration and feedback (Guo et al., 2025; Silver & Sutton, 2025). Within this paradigm, Reinforcement Learning with Verifiable Rewards (RLVR) (Wen et al., 2025) has become a particularly significant and efficient method. RLVR leverages outcome supervision—generating reward signals directly based on the correctness of the final answer—making it well-suited for domains like mathematics that have clear, verifiable solutions (Shao et al., 2024).

However, when the RLVR algorithm (e.g., GRPO (Shao et al., 2024)) maximizes rewards, it often leads to a decline in policy diversity and policy entropy collapse (Yu et al., 2025), limiting the model's long-term improvement. To break this performance plateau, recent research has focused on enhancing the model's exploration capabilities during training. Some works improve the optimization algorithm itself, aiming to sustain a higher policy entropy in the hope that the model will continuously explore more diverse reasoning paths (Yu et al., 2025; Cheng et al., 2025; Cui et al., 2025; Wang et al., 2025). A representative work in this direction, DAPO (Yu et al., 2025), introduces its Clip-Higher mechanism to prevent premature policy entropy collapse and sustain exploration. However, the primary goal of these methods is to increase the quantity of exploration or to maintain a higher entropy. As recent work has pointed out in Critique-GRPO (Zhang et al., 2025), higher entropy does not always guarantee efficient exploration; the quality of exploration can be more critical than its quantity. To pursue higher-quality exploration, another powerful line of research involves introducing external guidance. Representative works such as Critique-GRPO (Zhang et al., 2025) and LUFFY (Yan et al., 2025) leverage powerful external critique models (e.g., GPT-4o) or offline expert trajectories, respectively, to provide precise guidance signals, achieving powerful performance.

Despite significant progress, these approaches have not fully addressed a fundamental challenge in RLVR training: how to achieve efficient and targeted endogenous learning without relying on expensive external guidance (Schulman et al., 2017; Mnih et al., 2016).

To address this challenge, we propose **CLPO (Curriculum-guided Learning for Policy Optimization)**, a novel framework that actualizes a paradigm of **Guided Self-Evolution** (Silver et al., 2018). The core innovation of CLPO is its elevation of rollout information from a mere reward calculation signal to the central driver for constructing a dynamic curriculum. It employs **Online Curriculum Learning** to assess problem difficulty in real-time, which in turn guides an **Adaptive Problem Restructuring** mechanism to diversify medium problems and simplify hard ones. Furthermore, we introduce Difficulty-aware Policy Optimization, which integrates the curriculum signal into the optimization process via **dynamic KL regularization**. Through these mechanisms, CLPO transforms the training process into a structured, adaptive curriculum that co-evolves with the model's capabilities, all without any external dependencies.

We validate the effectiveness of CLPO on the Qwen3-8B (Yang et al., 2025) base model. Across eight mathematical and general reasoning benchmarks, including MATH-500 (Hendrycks et al., 2021), Minerva-Math (Lewkowycz et al., 2022), Olympiad (He et al., 2024), AMC23, AIME2024 (Li et al., 2024), TheoremQA (Chen et al., 2023), GPQA Diamond (Rein et al., 2024), and MMLU Pro (Wang et al., 2024), CLPO achieves an average **pass@1** performance improvement of **6.96%** over strong baselines. Our main contributions are as follows:

- We propose a novel framework to explore the previously unaddressed problem of inefficient learning arising from **uniform data sampling in RLVR**, effectively transforming unguided exploration into structured, purposeful self-improvement.

- We design a Difficulty-aware Policy Optimization mechanism that deeply integrates the curriculum signal with the policy optimization process via dynamic KL regularization to balance exploration and exploitation.

- We achieve state-of-the-art (SOTA) pass@1 performance on the highly challenging **AIME2024** benchmark, yielding a **3.3%** over the strongest baseline and validating the effectiveness of our method.

- We will open-source all our code and training scripts, hoping to inspire the community to further explore and utilize the rich information generated during the rollout phase.

## 2 RELATED WORK

### 2.1 ADAPTIVE METHODS IN ONLINE REINFORCEMENT LEARNING

Reinforcement Learning with Verifiable Rewards (RLVR) (Wen et al., 2025) has become a key technique for enhancing the reasoning abilities of LLMs. Foundational algorithms in this domain, such as PPO (Schulman et al., 2017) and GRPO (Shao et al., 2024), enable models to learn from their own generated experiences via online sampling and policy gradient updates. To further improve learning efficiency and exploration, a series of recent works have begun to explore more sophisticated uses of online information, developing diverse adaptive strategies. For instance, DAPO (Yu et al., 2025) and GFPO (Shrivastava et al., 2025) filter samples to focus on high-value data, while SVS (Liang et al., 2025) augments the problem set to maintain exploration diversity. Other approaches refine the optimization process itself; GSPO (Zheng et al., 2025) elevates rewards from the token to the sequence level, PKPO (Walder & Karkhanis, 2025) directly optimizes the pass@k metric to broaden the solution space. Our **CLPO** introduces a deeper, pedagogically-motivated adaptive paradigm. Through its Online Curriculum Learning mechanism, it ensures that every act of Adaptive Problem Restructuring is purposeful (simplification or diversification), aiming for higher-quality exploration.

### 2.2 EXTERNAL GUIDANCE IN ONLINE LEARNING

Distinct from improving exploration through internal adaptive mechanisms, another line of research enhances learning quality by introducing external guidance. Representative works in this area include LUFFY (Yan et al., 2025) and Critique-GRPO (Zhang et al., 2025). LUFFY incorporates an imitation learning objective on high-quality offline expert trajectories into its RL objective. Critique-GRPO leverages a powerful external critique model (e.g., GPT-4o) to generate fine-grained linguistic feedback that guides the model's refinement process online. Seeking even more granular feedback, other approaches such as Process Supervision (Cobbe et al., 2021) train a reward model to evaluate each intermediate step of the reasoning process, not just the final outcome. Our approach aligns with the goal of guided learning but differs fundamentally in its implementation. CLPO requires no external teacher models or offline expert data; its guidance signal is derived entirely from introspection on its own online performance, achieving a purely self-contained form of guidance.

### 2.3 CURRICULUM LEARNING FOR LARGE LANGUAGE MODELS

Curriculum Learning (CL) (Bengio et al., 2009) is a classic machine learning paradigm that mimics human learning by presenting training samples in an easy-to-hard order to accelerate convergence and improve final performance. In the field of natural language processing, applications of CL are often static and pre-defined, unable to adapt to the model's evolving capabilities during training. Some efforts have aimed to create dynamic curricula; for instance, FASTCURL (Song et al., 2025) employs a staged strategy that organizes the curriculum based on input prompt length while progressively scaling the context window. Other approaches, such as Train Long, Think Short (Hammoud et al., 2025), use a dynamically decaying token budget to guide length-controlled reasoning, enabling a smooth transition from exploring long trajectories to distilling concise solutions. Taking a step further, AdaRFT (Shi et al., 2025) introduces a reward-driven approach, adaptively adjusting the difficulty of training tasks based on the model's recent performance signals to keep it within

an optimal learning zone. Our CLPO brings the idea of curriculum learning into a new dimension, being the first to deeply integrate dynamic, online curriculum learning with reinforcement learning finetuning. It uses the model's real-time performance during rollouts to dynamically construct the curriculum, which in turn drives subsequent data restructuring and policy optimization, achieving truly adaptive learning.

# 3 METHODOLOGY

## 3.1 PRELIMINARY: GROUP RELATIVE POLICY OPTIMIZATION

The central innovation of Group Relative Policy Optimization (GRPO) (Guo et al., 2025) lies in its critic-free approach to advantage estimation. Specifically, for a given prompt, the algorithm normalizes the reward of each generated response against the mean and standard deviation of rewards from a concurrently sampled group of its peers. The process involves sampling $G$ responses $\{y_i\}_{i=1}^{G}$ for a question $q$ from an old policy $\pi_{\theta_{\text{old}}}$, and assigning a reward $r_i$ to each. The GRPO training objective is formulated as:

$$\mathcal{J}_{\text{GRPO}}(\theta) = \mathbb{E}_{(q,a)\sim\mathcal{D},\{y_i\}_{i=1}^{G}\sim\pi_{\theta_{\text{old}}}(\cdot|q)}$$

$$\left[ \frac{1}{G}\sum_{i=1}^{G}\frac{1}{|y_i|}\sum_{t=1}^{|y_i|}\Big(\min(k_{i,t}(\theta)\hat{A}_{i,t}, \text{clip}(k_{i,t}(\theta), 1-\epsilon, 1+\epsilon)\hat{A}_{i,t}) - \beta D_{\text{KL}}(\pi_\theta\|\pi_{\text{ref}})\Big) \right] \quad (1)$$

where $k_{i,t}(\theta) = \frac{\pi_\theta(y_{i,t}|q,y_{i,<t})}{\pi_{\theta_{\text{old}}}(y_{i,t}|q,y_{i,<t})}$ and $\hat{A}_{i,t} = \frac{r_i - \text{mean}(\{r_j\}_{j=1}^{G})}{\text{std}(\{r_j\}_{j=1}^{G})}$. The GRPO algorithm utilizes this group-relative advantage within its clipped objective function to achieve stable and efficient policy optimization.

## 3.2 CLPO: AN OVERVIEW

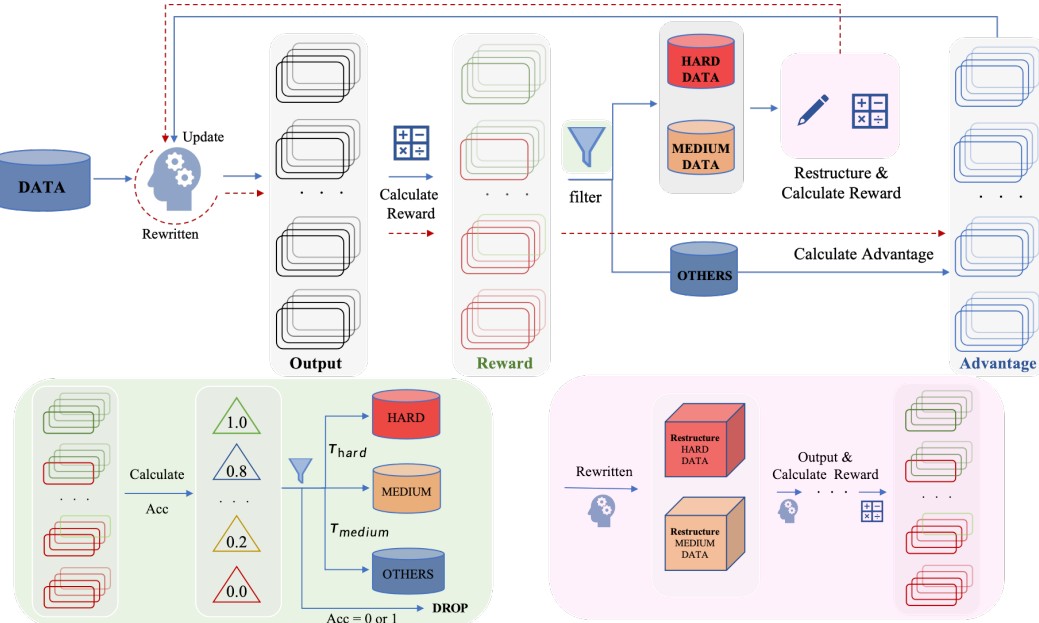

Figure 2: The overall workflow of CLPO. The framework consists of three main stages detailed in the green and pink boxes: (1) Online Curriculum Learning, (2) Adaptive Problem Restructuring, and (3) Value-Driven Filtering and Optimization.

While standard RLVR methods like GRPO operate on a static dataset, CLPO introduces a paradigm of Guided Self-Evolution. As illustrated in Figure 2, at each training step, CLPO dynamically reconstructs the training batch $B_{\text{mix}}$ to create a valuable mixture of problems tailored to the model's

current ability. The optimization phase of CLPO then proceeds by maximizing our objective over this dynamic batch. This objective estimates advantages using group-relative performance and is regularized by a difficulty-aware KL penalty:

$$J_{\text{CLPO}}(\theta) = \mathbb{E}_{(q,a)\sim B_{\text{mix}},\{y_i\}_{i=1}^{G}\sim \pi_{\theta_{\text{old}}}(\cdot|q)}$$

$$\left[\frac{1}{\sum_{i=1}^{G}|y_i|}\sum_{i=1}^{G}\sum_{t=1}^{|y_i|}\left\{\min(k_{i,t}(\theta)\hat{A}_{i,t}, \text{clip}(k_{i,t}(\theta), 1-\epsilon, 1+\epsilon)\hat{A}_{i,t}) - \lambda_d\beta \cdot D_{\text{KL}}(\pi_\theta\|\pi_{\text{ref}})\right\}\right] \quad (2)$$

The effectiveness of CLPO stems from three core mechanisms, which we detail in the following sections: Online Curriculum Learning, Adaptive Problem Restructuring, and Difficulty-aware Policy Optimization.

### 3.3 ONLINE CURRICULUM LEARNING

The foundational mechanism of CLPO is **Online Curriculum Learning**, which transforms the rollout phase from a mere data collection step into a real-time diagnostic tool. In each training step, for every problem $q$ in the sampled batch $\mathcal{D}_{\text{batch}}$, we generate $G$ responses $\{y_i\}_{i=1}^{G}$ using the current policy $\pi_\theta$. We then compute the empirical accuracy $\text{Acc}(q, \pi_\theta)$ using a **Verifier**:

$$\text{Acc}(q, \pi_\theta) = \frac{1}{G}\sum_{i=1}^{G}\mathbb{I}(\text{Verifier}(y_i, a) = \text{True}) \quad (3)$$

This accuracy score serves as a direct, online measure of the model's current proficiency on the problem. Based on this, we utilize predefined difficulty thresholds $(\tau_{\text{hard}}, \tau_{\text{med}})$ to dynamically partition the batch into two key subsets for subsequent action:

$$\mathcal{D}_{\text{batch}}^{\text{hard}} = \{q \in \mathcal{D}_{\text{batch}} \mid \text{Acc}(q, \pi_\theta) \leq \tau_{\text{hard}}\} \quad (4)$$

$$\mathcal{D}_{\text{batch}}^{\text{med}} = \{q \in \mathcal{D}_{\text{batch}} \mid \tau_{\text{hard}} < \text{Acc}(q, \pi_\theta) \leq \tau_{\text{med}}\} \quad (5)$$

These sets, the **batch hard set** and the **batch medium set**, provide the precise, real-time guidance signal for the next stage of our framework.

### 3.4 ADAPTIVE PROBLEM RESTRUCTURING

To achieve high-quality, guided exploration, CLPO introduces **Adaptive Problem Restructuring (APR)**, a mechanism where the model acts as its own teacher to create more suitable learning materials. This mechanism is implemented through a restructuring function $f_p(q, d)$, which leverages two distinct **Prompts** ($p$) tailored to the problem's difficulty $d$ (see AppendixE for the specific prompts):

$$q' = f_p(q, d), \quad \text{where } d \in \{\text{hard}, \text{medium}\} \quad (6)$$

The restructuring strategy is guided by the online curriculum. For problems classified as medium ($d = \text{medium}$), we employ a **Diversification** prompt. This prompt aims to enhance the model's robustness by generating semantically equivalent but varied phrasings, encouraging it to grasp deeper semantic commonalities. For problems classified as hard ($d = \text{hard}$), we utilize a **Simplification** prompt. Its objective is to generate a restructured problem $q'$ that is easier for the model to comprehend, transforming an initially intractable problem into an effective and learnable training signal. In both cases, the original answer $a$ is strictly preserved.

### 3.5 VALUE-DRIVEN FILTERING AND DIFFICULTY-AWARE OPTIMIZATION

In order to ensure maximum training efficiency, CLPO employs a two-stage Value-Driven Filtering process to construct the final training batch, followed by Difficulty-aware Policy Optimization.

First, we filter the original batch to form the **base set** $B_{\text{base}}$, retaining only the problems within the model's learning area (i.e., accuracy $0 < \text{Acc}(q, \pi_\theta) < 1$):

$$B_{\text{base}} = \{(q, a) \mid q \in \mathcal{D}_{\text{batch}}, 0 < \text{Acc}(q, \pi_\theta) < 1\} \quad (7)$$

Second, the newly restructured problems $q'$ undergo the same filtering criterion, forming the Restructure set $B_{\text{restructure}}$:

$$B_{\text{restructure}} = \{(q', a) \mid q' = f_p(q, d), q \in \mathcal{D}_{\text{batch}}^{\text{hard}} \cup \mathcal{D}_{\text{batch}}^{\text{med}}, 0 < \text{Acc}(q', \pi_\theta) < 1\} \qquad (8)$$

The final training batch is the union $B_{\text{mix}} = B_{\text{base}} \cup B_{\text{restructure}}$. When optimizing on this high-quality data, we employ Difficulty-aware Policy Optimization. This is achieved via a dynamic KL regularization mechanism, where the scaling factor $\lambda_d$ in Eq. equation 2 depends on the problem's original difficulty. By setting $\lambda_{\text{hard}} < \lambda_{\text{non-hard}}$, we encourage greater policy exploration on difficult problems while enforcing stronger regularization on problems where the model already has a foothold, thereby achieving a dynamic balance between exploration and exploitation.

### 3.6 ALGORITHM

We integrate the core mechanisms into the CLPO algorithm. At each training step, the algorithm first performs an online evaluation on a sampled batch to determine the difficulty level of each problem. Specifically, it generates multiple candidate solutions using the current policy to calculate an empirical accuracy score, which is then used to categorize problems as "hard" or "medium". Subsequently, the algorithm applies adaptive problem restructuring to these identified problems, performing simplification and diversification, respectively. Finally, a rigorous value-driven filter is applied to all original and restructured problems, and only the high-value samples with an accuracy strictly between 0 and 1 are combined to form the final training batch, $B_{\text{mix}}$, for the policy update. The complete pseudocode for this algorithm, alongside a deeper theoretical analysis and implementation details for each mechanism, is provided in the Appendix 1.

## 4 EXPERIMENT

To systematically evaluate the effectiveness of our proposed CLPO algorithm, we conducted a series of comprehensive experiments. Our evaluation begins by comparing the performance of CLPO against current state-of-the-art finetuning methods on a wide range of mathematical reasoning benchmarks. Subsequently, we perform a series of detailed ablation studies to analyze the contribution of each core component within CLPO. Finally, we analyze the computational scaling properties of models trained with CLPO at test time.

### 4.1 EXPERIMENTAL SETUP

We select **Qwen3-8B** (Yang et al., 2025) as the base model for all experiments and conduct training on the **DAPO-Math-17k** (Yu et al., 2025) dataset. To comprehensively evaluate model performance, we employ a diverse evaluation suite comprising eight benchmarks, which can be categorized into two main groups. The **In-Domain** test sets, which are stylistically and topically closer to the training data, include MATH 500, Minerva MATH, Olympiad Bench, AMC23, and AIME24 (Hendrycks et al., 2021; Lewkowycz et al., 2022; He et al., 2024; Li et al., 2024). The **Out-of-Distribution (OOD)** test sets, used to assess the model's generalization and reasoning capabilities in different domains, include TheoremQA, GPQA Diamond, and MMLU Pro (Chen et al., 2023; Rein et al., 2024; Wang et al., 2024).

All our experiments were conducted on 8x H20 GPUs. The models were trained for 200 steps on the DAPO-Math-17k dataset, and the checkpoint with the best performance on a validation set was selected for final evaluation. During training, the rollout number was set to n=4 for each prompt, with a decoding temperature of 1.0 and a maximum response length of 8192 tokens. For CLPO, we set the default difficulty thresholds to **(0.3, 0.7)** and the dynamic KL regularization scaling factors to $(\lambda_{\text{hard}}, \lambda_{\text{non-hard}}) = (0.3, 1.0)$. We used a constant learning rate schedule with a learning rate of 1e-6. For our main experiments, we report pass@1 as the primary evaluation metric. All ablation studies were conducted on the AIME24 test set to analyze the impact of each component on challenging problems.

### 4.2 MAIN RESULTS

We compared the performance of CLPO with all baseline methods across the in-domain and out-of-distribution test sets, with detailed results presented in Table 1. The results clearly demonstrate

Table 1: Performance of CLPO against state-of-the-art finetuning methods on Qwen3-8B. Our method, CLPO, operating under the *Guided Self-Evolution* paradigm, demonstrates superior performance across all benchmarks without relying on external guidance.

| Method | Optimization Policy | Math (In-Domain) | | | | | General Reasoning (OOD) | | | Avg |
|---|---|---|---|---|---|---|---|---|---|---|
| | | MATH 500 | Minerva Math | Olympiad Bench | AMC23 | AIME24 | MMLU Pro | Theorem QA | GPQA Diamond | |
| *All methods are finetuned on Qwen3-8B* | | | | | | | | | | |
| *Supervised Fine-Tuning (SFT)* | | | | | | | | | | |
| RAFT | Ranking-Based Imitation | 76.20 | 35.58 | 36.86 | 50.00 | 26.67 | 65.93 | 43.50 | 36.76 | 46.44 |
| Refinement FT | Guided Refinement | 83.20 | 47.58 | 40.71 | 70.00 | 33.33 | 67.84 | 41.29 | 34.47 | 52.30 |
| Critique FT | Learning to Critique | 79.00 | 35.23 | 39.64 | 67.50 | 33.33 | 63.16 | 46.00 | 34.84 | 49.84 |
| CITL-FT | Mixed-Data SFT | 76.40 | 37.20 | 38.57 | 62.50 | 30.00 | 66.13 | 44.25 | 36.36 | 48.93 |
| *Reinforcement Learning with Verifiable Rewards (RLVR)* | | | | | | | | | | |
| GRPO | Group-Based RL | 89.20 | 51.47 | 57.40 | 82.50 | 43.33 | 69.86 | 54.75 | 47.80 | 62.04 |
| DAPO | Dynamic Sampling | **91.20** | 53.31 | 63.80 | 87.50 | 46.67 | 70.01 | 55.00 | 48.48 | 64.50 |
| LUFFY | Off-Policy Imitation | 89.40 | 52.94 | 58.80 | 85.00 | 40.00 | 70.34 | 58.25 | 49.49 | 63.03 |
| Critique-GRPO (Simple) | Critique-Driven RL | 89.40 | 52.57 | 60.20 | 87.50 | 40.00 | 70.13 | 59.00 | 48.63 | 63.43 |
| Critique-GRPO (CoT) | Critique-Driven RL | **91.20** | 61.50 | 63.80 | **90.00** | 46.67 | 70.98 | 59.50 | 50.50 | 66.77 |
| **CLPO (Ours)** | **Guided Self-Evolution** | 89.60 | **76.10** | **77.50** | **90.00** | **50.00** | **72.39** | **71.63** | **62.63** | **73.73** |

that CLPO achieves state-of-the-art performance across most of eight benchmarks. It is particularly noteworthy that even on the most challenging in-domain datasets (e.g., Olympiad Bench, AIME24) and all OOD datasets, CLPO shows significant improvements over the powerful methods, such as Critique-GRPO. This outcome strongly suggests that our proposed framework of online curriculum learning and adaptive problem restructuring effectively enhances the model's deep reasoning and generalization capabilities, rather than merely memorizing patterns in the training data.

## 4.3 ABLATION STUDIES

To deeply investigate the effectiveness of each core design module within CLPO, we conducted a series of detailed ablation studies on the challenging **AIME24** dataset. All results from our ablation studies are presented as **Avg@32** to better evaluate the upper bound of the model's capabilities and the diversity of its exploration.

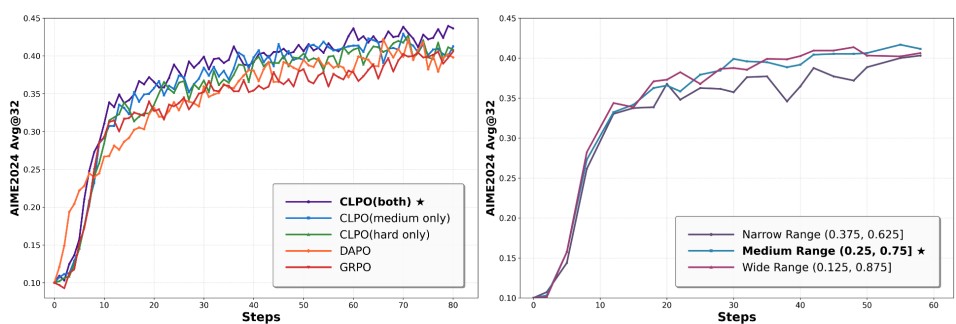

Figure 3: Ablation studies on Adaptive Problem Restructuring. Left: Comparison of restructuring strategies. Right: Impact of difficulty ranges. All experiments sampled each problem 8 times.

**Impact of Adaptive Problem Restructuring.** We first analyze the impact of different strategies and parameters within our **Adaptive Problem Restructuring (APR)** mechanism, with all experiments conducted by sampling each problem 8 times and using a default medium difficulty range of (0.25, 0.75). The left panel of Figure 3 compares different restructuring strategies. We observe that diversifying medium-difficulty problems only (`medium only`) yields better performance than simplifying hard problems only (`hard only`). We attribute this to the fact that problems classified as medium lie at the edge of the model's current cognitive learning zone, and diversifying

them is the most effective way to consolidate existing knowledge and generalize to new domains. Nonetheless, the strategy of only restructuring hard problems still outperforms baselines like GRPO and DAPO. This is thanks to our dynamic training set and the guided self-evolution the model undergoes via simplification, which allows it to extract effective signals from previously intractable problems. Ultimately, the results clearly show that the full CLPO strategy (`both`), which combines both approaches, achieves the best performance. This demonstrates that our framework perfectly synergizes two learning modes: it both expands generalization capabilities on moderately mastered problems and enables effective learning through the simplification strategy on challenging ones, leading to the most comprehensive capability improvement.

The right panel of Figure 3 investigates the choice of the difficulty range in **Online Curriculum Learning**. We find that a moderately sized range is crucial. If the range is too wide (e.g., (0.125, 0.875]), some problems that are inherently very difficult for the model are misclassified as "medium" and subjected to diversification. This process is tantamount to turning one hard problem into another equally difficult variation, leading to inefficient learning. At the same time, overly simple problems may also be included for restructuring, causing the model to waste computational resources on low-information-gain samples. Conversely, if the range is too narrow (e.g., (0.375, 0.625]), a large number of valuable problems at the model's capability frontier are excluded from the restructuring process. This results in insufficient raw material for the curriculum, thereby limiting the algorithm's optimization potential. Therefore, a medium-sized range, such as (0.25, 0.75], most accurately filters for the candidate problems that are most valuable for the model's current stage, striking an optimal balance between learning efficiency and full data utilization.

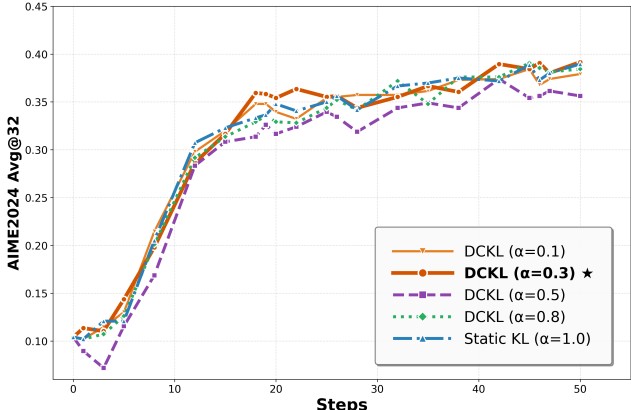

Figure 4: Performance comparison of dynamic KL regularization with different scaling factors ($\alpha$) against a static KL baseline.

**Role of Difficulty-aware Policy Optimization.**   Finally, we analyze our **Difficulty-aware Policy Optimization** mechanism, specifically the effectiveness of dynamic KL regularization (DCKL). In Figure 4, we compare the performance of dckl with different scaling factors $\alpha$ (i.e., $\lambda_{hard}$) against a static KL baseline ($\alpha = 1.0$), where a lower $\alpha$ represents a weaker KL constraint on difficult problems. The experimental results clearly reveal the critical importance of striking a precise balance between exploration and exploitation. We observe the worst performance when $\alpha$ is set to 0.5. We infer that this value falls into a suboptimal exploration range: the policy constraint is strong enough to hinder truly novel solution attempts, yet weak enough to disrupt policy stability. Interestingly, the setting with a weak policy constraint ($\alpha = 0.1$) and those with strong policy constraints ($\alpha = 0.8$ and the static KL, $\alpha = 1.0$) ultimately converge to a similar level of suboptimal performance. The former's convergence stability may be compromised by insufficient constraint, while the latter are too restrictive to effectively learn new solutions for difficult problems. In contrast, setting $\alpha$ to **0.3** strikes the optimal balance between exploration freedom and policy stability, achieving the best performance. CLPO, by precisely adjusting the policy constraint for difficult problems, provides effective guidance for high-quality exploration and thus surpasses other static or suboptimal optimization strategies.

## 4.4 Test-Time Scaling Analysis

Finally, we investigate the test-time scaling of the model trained with CLPO, specifically its **pass@k** performance. This metric measures the probability of generating at least one correct answer in k independent samples and is widely regarded as a key indicator of a model's **problem-solving ceiling** and its **coverage of the valid solution space**. We evaluate pass@k by generating k solutions for each problem at test time, where k varies from $2^0 = 1$ to $2^6 = 64$. We compare CLPO against the base model (Qwen3-8B), GRPO, and DAPO on the **AIME2024** and **AIME2025** test sets, with the results plotted in Figure 5.

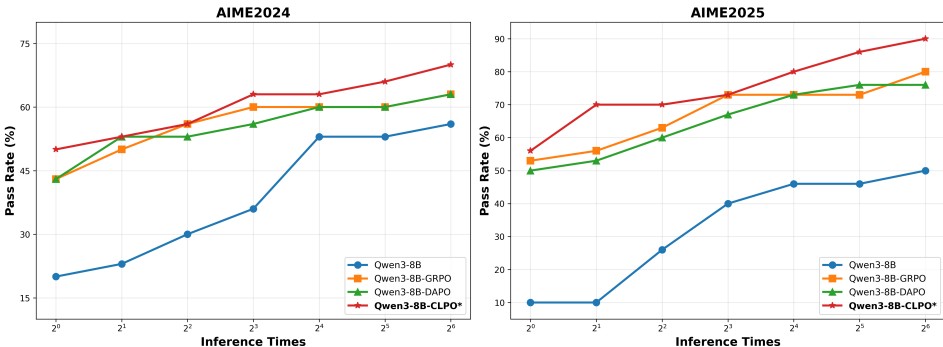

Figure 5: Test-time scaling performance (pass@k) on AIME2024 (left) and AIME2025 (right). The pass rate is evaluated over $k$ inference times, where $k$ ranges from 1 to 64. CLPO outperforms the baselines across all sampling counts.

Two phenomena can be observed from the plots. First, at all sample counts k, the pass rate (pass@k) of CLPO is **higher** than all baseline methods. This demonstrates that the model trained with CLPO not only has a higher single-sample success rate but also explores a broader valid solution space. This suggests that the solutions generated by the CLPO-trained model are not only of **higher individual correctness** but also richer in **solution path diversity**, allowing it to more efficiently cover a correct answer through multiple samples.

## 5 Conclusion

In this paper, we proposed CLPO, a novel framework for guided self-evolution designed to address the inefficient learning caused by the uniform training paradigm in existing RLVR methods. The core innovation of this approach is its elevation of online performance signals from the rollout phase to become the central driver for constructing a dynamic curriculum. Through the synergy of three core mechanisms—Online Curriculum Learning, Adaptive Problem Restructuring, and Difficulty-aware Policy Optimization—our framework transforms the training process into an adaptive pedagogical loop that co-evolves with the model's capabilities, all without any external dependencies. Our extensive experiments on multiple challenging mathematical and general reasoning benchmarks demonstrate that this method significantly outperforms state-of-the-art finetuning approaches, achieving SOTA performance.

Despite the encouraging results, there are several avenues for future exploration. First, our current approach to assessing problem difficulty relies primarily on the final answer's correctness; future work could explore incorporating more fine-grained process signals to enable even more precise curriculum construction. Second, while this work primarily validates our method in the domain of mathematical reasoning, extending this dynamic learning paradigm to other complex reasoning domains, such as code generation and scientific question-answering, presents a promising direction. Finally, we believe that the adaptive learning capabilities championed by our framework will exhibit even greater potential as training data and model parameter scales continue to grow, and we plan to further validate its scalability on larger-scale models and datasets.

REPRODUCIBILITY STATEMENT

We provide a comprehensive set of resources to ensure the reproducibility of our work. Our algorithm, CLPO, is detailed in Appendix C and Algorithm 1. All hyperparameters for both the main training process and our CLPO-specific components are listed in Appendix B. To facilitate full replication of our results, we have included our complete, anonymized source code and scripts as part of the supplementary material. Furthermore, the appendix includes visualizations of the curriculum dynamics (Appendix D), the exact prompts used for problem restructuring (Appendix E), and detailed case studies (Appendix F) to provide deeper insights into our method's behavior.

ETHICS STATEMENT

This research was conducted with full consideration of and adherence to the ICLR Code of Ethics throughout its design, implementation, and reporting. Our work aims to advance the scientific understanding of how to enhance the reasoning capabilities of Large Language Models by proposing a novel, adaptive learning algorithm (CLPO). This goal aligns with the principles of contributing to society and upholding high standards of scientific excellence.

To ensure the transparency and reproducibility of our research, we have provided comprehensive implementation details, hyperparameters, and the exact prompts used for problem restructuring in the Appendix. Furthermore, our complete source code has been submitted as supplementary material to facilitate verification and further research by the community.

All datasets used in this study (e.g., DAPO-Math-17k) are publicly available academic benchmarks and do not involve any private or sensitive information, thus posing no privacy concerns. Our method is a general-purpose learning algorithm, and we do not foresee any direct, predictable negative societal impacts or potential harms arising from our work. We have also explicitly disclosed our use of Large Language Models as writing assistants in the preparation of this manuscript in the Appendix.

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

# APPENDIX

## A    USAGE OF LARGE LANGUAGE MODELS

During the preparation of this manuscript, we utilized Large Language Models (LLMs) as a writing assistant. The usage of LLMs was strictly limited to improving the fluency, clarity, and grammatical correctness of the language, such as rephrasing sentences or correcting grammatical errors. LLMs were not involved in the core research ideation, experimental design, analysis of results, or the formulation of conclusions presented in this paper.

## B    IMPLEMENTATION DETAILS

Table 2: Main hyperparameters for training, evaluation, and environment.

| Name | Value | Description |
|---|---|---|
| *Training* | | |
| Base Model | Qwen3-8B | The base model used in experiments. |
| Dataset | DAPO-Math-17k | The dataset used for training. |
| training_steps | 200 | Total number of training steps. |
| Optimizer | AdamW | The optimizer used. |
| lr | 1e-6 (Constant) | Learning rate. |
| batch_size | 64 | Global batch size during training. |
| n_rollouts ($G$) | 4 | Number of rollouts per prompt. |
| rewards | 1 or -1 | Scalar rewards for correct/incorrect responses. |
| kl_loss_coef ($\beta$) | 0.001 | Coefficient for KL divergence loss. |
| grad_clip | 1.0 | Gradient clipping threshold. |
| train_temp | 1.0 | Sampling temperature during training rollout. |
| top_p | 1.0 | Top-p sampling parameter during training rollout. |
| max_response_length | 8192 | Maximum length of generated responses. |
| *Evaluation* | | |
| val_temp | 1.0 | Sampling temperature during evaluation. |
| *Environment* | | |
| Hardware | $8 \times$ NVIDIA H20 | Hardware used for experiments. |
| Software | verl (Sheng et al., 2025) | Software frameworks used. |

Table 3: CLPO-specific hyperparameters.

| Name | Value | Description |
|---|---|---|
| hard_threshold ($\tau_{\text{hard}}$) | 0.3 | Accuracy threshold for hard problems. |
| medium_threshold ($\tau_{\text{med}}$) | 0.7 | Accuracy threshold for medium problems. |
| hard_kl_scaler ($\lambda_{\text{hard}}$) | 0.3 | KL scaling factor for hard problems. |
| non_hard_kl_scaler ($\lambda_{\text{non-hard}}$) | 1.0 | KL scaling factor for non-hard problems. |

This section provides the detailed experimental setup to ensure the reproducibility of our work. All our experiments were conducted using the **vLLM** inference framework (Kwon et al., 2023). Table 2 lists the main hyperparameters for our experiments, and Table 3 details the parameters specific to CLPO.

## C    THEORETICAL COMPARISON BETWEEN CLPO AND GRPO

This section provides a more formal and detailed analysis of the key theoretical differences between the learning paradigms of standard GRPO and our proposed CLPO.

## C.1 GRPO: A STATIC LEARNING PARADIGM

First, we recall the objective function of GRPO from Eq. (1):

$$\mathcal{J}_{\text{GRPO}}(\theta) = \mathbb{E}_{(q,a)\sim\mathcal{D},\{y_i\}_{i=1}^{G}\sim\pi_{\theta_{\text{old}}}(\cdot|q)}$$

$$\left[\frac{1}{G}\sum_{i=1}^{G}\frac{1}{|y_i|}\sum_{t=1}^{|y_i|}\left(\min(k_{i,t}(\theta)\hat{A}_{i,t}, \text{clip}(k_{i,t}(\theta), 1-\epsilon, 1+\epsilon)\hat{A}_{i,t}) - \beta D_{\text{KL}}(\pi_\theta\|\pi_{\text{ref}})\right)\right] \quad (9)$$

The core of GRPO lies in its advantage estimation $\hat{A}_{i,t}$, which is entirely dependent on the rewards of responses $\{y_i\}$ generated for a *given* question $q$. The expectation $\mathbb{E}_{(q,a)\sim\mathcal{D}}$ reveals that the optimization is performed over a **static, policy-independent data distribution** $\mathcal{D}$. In each training step, the sampling probability of a problem $q$, $P(q)$, remains constant, typically following a uniform distribution $P(q) = 1/|\mathcal{D}|$.

This implies that the learning process of GRPO follows a **static paradigm**. Under this paradigm, the model passively receives problems sampled from $\mathcal{D}$, and its only agency lies in optimizing its **response space** $\pi(y|q)$ to assign higher probabilities to "good" responses. However, the model cannot actively influence the **problem space** it is exposed to. This uniform sampling strategy presents two main theoretical limitations. First, when the policy $\pi_\theta$ has already achieved high proficiency on a subset of $\mathcal{D}$ (e.g., easy problems), GRPO continues to allocate the same computational resources to sampling and exploring these problems. This leads to sparse learning signals, as the rewards for all responses on these mastered problems may be very similar, causing the advantage $\hat{A}_{i,t}$ to approach zero and thus providing no effective gradient for policy updates. Second, for problems that are too difficult for the current model, all responses may receive zero or negative rewards, which can also lead to the advantage signal within a group collapsing to zero, trapping the model in a state of "persistent failure" without making progress. Both scenarios result in wasted computational resources and the emergence of performance plateaus.

## C.2 CLPO: A DYNAMIC LEARNING CURRICULUM

CLPO fundamentally alters this static learning paradigm by introducing a dynamic curriculum that co-evolves with the model's capabilities. We recall the CLPO objective from Eq. (2):

$$J_{\text{CLPO}}(\theta) = \mathbb{E}_{(q,a)\sim B_{\text{mix}},\{y_i\}_{i=1}^{G}\sim\pi_{\theta_{\text{old}}}(\cdot|q)}$$

$$\left[\frac{1}{\sum_{i=1}^{G}|y_i|}\sum_{i=1}^{G}\sum_{t=1}^{|y_i|}\left\{\min(k_{i,t}(\theta)\hat{A}_{i,t}, \text{clip}(k_{i,t}(\theta), 1-\epsilon, 1+\epsilon)\hat{A}_{i,t}) - \lambda_d\beta \cdot D_{\text{KL}}(\pi_\theta\|\pi_{\text{ref}})\right\}\right] \quad (10)$$

The core improvements here are twofold: a **dynamic training data distribution** and a **dynamic regularization strength**.

### C.2.1 DYNAMIC TRAINING DATA DISTRIBUTION

This is the most central theoretical innovation of CLPO. The key difference lies in the expectation $\mathbb{E}_{(q,a)\sim B_{\text{mix}}}$. The training batch $B_{\text{mix}}$ is not directly sampled from the static distribution $\mathcal{D}$, but is dynamically constructed via a deterministic function $g(\cdot)$ that depends on the performance of the **current policy** $\pi_\theta$:

$$B_{\text{mix}} = g(\mathcal{D}_{\text{batch}}, \pi_\theta) \quad (11)$$

This function $g$ corresponds to our Online Curriculum Learning and Adaptive Problem Restructuring mechanisms. We can formalize this process in three steps.

**Step 1: Online Curriculum Learning and Partitioning.** The Online Curriculum Learning mechanism first partitions the original batch $\mathcal{D}_{\text{batch}}$ into three disjoint subsets based on the online accuracy of $\pi_\theta$, $\text{Acc}(q, \pi_\theta)$: the base training set $\mathcal{D}_{\text{base}}$, the restructuring candidates $\mathcal{D}_{\text{candidates}}$, and the dropped set $\mathcal{D}_{\text{drop}}$.

$$\mathcal{D}_{\text{base}} = \{(q,a) \in \mathcal{D}_{\text{batch}} \mid \tau_{\text{med}} < \text{Acc}(q, \pi_\theta) < 1\} \quad (12)$$

$$\mathcal{D}_{\text{candidates}} = \{(q,a) \in \mathcal{D}_{\text{batch}} \mid 0 < \text{Acc}(q, \pi_\theta) \le \tau_{\text{med}}\} \quad (13)$$

$$\mathcal{D}_{\text{drop}} = \{(q,a) \in \mathcal{D}_{\text{batch}} \mid \text{Acc}(q, \pi_\theta) \in \{0, 1\}\} \quad (14)$$

Samples in $\mathcal{D}_{\text{base}}$ are deemed suitably challenging and will directly be included in the final training batch.

**Step 2: Adaptive Problem Restructuring and Filtering.**   The Adaptive Problem Restructuring mechanism, $f_p(q, d)$, is applied to each problem $q$ in $\mathcal{D}_{\text{candidates}}$. For each candidate, its difficulty level $d$ (hard or medium) is determined, and its restructured version $q'$ is generated:

$$\forall (q, a) \in \mathcal{D}_{\text{candidates}}, \quad q' = f_p(q, \text{GetDifficulty}(\text{Acc}(q, \pi_\theta))) \tag{15}$$

All generated problems $\{q'\}$ form a temporary set $\mathcal{D}'_{\text{temp}}$. This temporary set then undergoes a second round of **value filtering**, where only the problems that remain within the model's effective learning zone post-restructuring are retained. This forms the final restructured training set, $\mathcal{D}_{\text{restructured}}$.

$$\mathcal{D}_{\text{restructured}} = \{(q', a) \mid (q, a) \in \mathcal{D}_{\text{candidates}}, q' = f_p(q, d), 0 < \text{Acc}(q', \pi_\theta) < 1\} \tag{16}$$

**Step 3: Final Training Batch Construction.**   The final dynamic training batch $B_{\text{mix}}$ is the union of the base training set and the doubly-filtered restructured set:

$$B_{\text{mix}} = \mathcal{D}_{\text{base}} \cup \mathcal{D}_{\text{restructured}} \tag{17}$$

This rigorous, policy-dependent procedure implies that the probability of a problem instance appearing in the final training batch is no longer its intrinsic probability $P(q)$, but a complex **conditional probability that depends on** $\pi_\theta$. Therefore, the optimization in CLPO is effectively performed over a **dynamic, policy-dependent effective data distribution** $P_{\text{eff}}(q'|\pi_\theta)$.

### C.2.2 Dynamic Regularization: From Global Compromise to Sample-level Adaptive Optimization

Beyond the dynamic data distribution, the second core theoretical advantage of CLPO lies in its **Difficulty-aware Policy Optimization**, actualized through the dynamic KL regularization mechanism. The GRPO objective contains a **static, globally shared** KL penalty coefficient $\beta$. This fixed $\beta$ must strike a **global, suboptimal compromise** between exploration (which requires a smaller $\beta$ to allow policy deviation) and exploitation (which requires a larger $\beta$ to ensure stability). The same policy constraint is applied to all samples throughout training, regardless of their difficulty, which is theoretically inefficient.

CLPO transforms this fixed KL penalty into a **dynamic, sample-level policy constraint** by introducing the difficulty-aware scaling factor $\lambda_d$. We can define an **effective KL coefficient**, $\beta_{\text{eff}}$:

$$\beta_{\text{eff}}(q, \pi_\theta) = \lambda_d \cdot \beta = \begin{cases} \lambda_{\text{hard}} \cdot \beta & \text{if Acc}(q, \pi_\theta) \leq \tau_{\text{hard}} \\ \lambda_{\text{non-hard}} \cdot \beta & \text{otherwise} \end{cases} \tag{18}$$

where we set $\lambda_{\text{hard}} < \lambda_{\text{non-hard}}$ (e.g., 0.3 vs. 1.0). This design has two distinct effects on the optimization process:

1. **For Hard Problems** ($\text{Acc}(q) \leq \tau_{\text{hard}}$)**:** In this case, $\beta_{\text{eff}}$ is small. From a policy optimization perspective, this is equivalent to **significantly widening the trust region** for this specific sample. For a difficult problem, the correct reasoning path may lie in a low-probability region of the current policy's distribution, far from the reference policy $\pi_{\text{ref}}$. A smaller KL penalty, $-\lambda_{\text{hard}}\beta D_{\text{KL}}(\pi_\theta \| \pi_{\text{ref}})$, allows for larger, more exploratory gradient steps during the policy update, increasing the likelihood that the model discovers these novel, high-reward reasoning trajectories.

2. **For Non-hard Problems:** In this case, $\beta_{\text{eff}}$ is large. This is equivalent to **tightening the trust region** for this sample. For problems that the model has partially mastered, the goal is stable and fine-grained refinement rather than drastic policy shifts. A larger KL penalty, $-\lambda_{\text{non-hard}}\beta D_{\text{KL}}(\pi_\theta \| \pi_{\text{ref}})$, constrains the policy update to stay in the vicinity of the reference policy, effectively preventing **catastrophic forgetting** of already-learned capabilities and ensuring the stability of the learning process.

**In summary**, this dual shift—from optimizing over a **fixed distribution** $P(q)$ **and a fixed regularization** to a **dynamic distribution** $P(q'|\pi_\theta)$ **and a dynamic regularization** $\lambda_d$—constitutes the

most fundamental theoretical advantage of CLPO over GRPO. It elevates the model from a passive problem-solver to an active curriculum-designer, enabling it to continuously focus both its **learning resources** and its **exploration intensity** on the frontier of its own capabilities, thereby breaking the bottlenecks of static learning and achieving more efficient and sustained self-evolution.

### C.3 ALGORITHM

---
**Algorithm 1** Curriculum-guided Learning for Policy Optimization (CLPO)

---
1: **Input:** Data $\mathcal{D}$, policy $\pi_\theta$, ref policy $\pi_{\text{ref}}$, thresholds $(\tau_{\text{hard}}, \tau_{\text{med}})$, rewrite func. $f_p(\cdot, d)$, samples $G$.
2: **for** each training step **do**
3:     Sample batch $\mathcal{D}_{\text{batch}} \subset \mathcal{D}$; Initialize $B_{\text{mix}} \leftarrow \emptyset$, $D_{\text{candidates}} \leftarrow \emptyset$.
4:     **for** $(q, a) \in \mathcal{D}_{\text{batch}}$ **do**
5:        $\text{Acc}(q) \leftarrow \text{EvaluateAccuracy}(\pi_\theta, q, a, G)$.
6:        **if** $0 < \text{Acc}(q) < 1$ **then**
7:           $B_{\text{mix}} \leftarrow B_{\text{mix}} \cup \{(q, a)\}$.
8:           **if** $\text{Acc}(q) \le \tau_{\text{med}}$ **then**
9:              $D_{\text{candidates}} \leftarrow D_{\text{candidates}} \cup \{(q, a)\}$.
10:           **end if**
11:        **end if**
12:     **end for**
13:     **for** $(q, a) \in D_{\text{candidates}}$ **do**
14:        $d \leftarrow \text{GetDifficulty}(\text{Acc}(q))$; $q' \leftarrow f_p(q, d)$.
15:        $\text{Acc}(q') \leftarrow \text{EvaluateAccuracy}(\pi_\theta, q', a, G)$.
16:        **if** $0 < \text{Acc}(q') < 1$ **then**
17:           $B_{\text{mix}} \leftarrow B_{\text{mix}} \cup \{(q', a)\}$.
18:        **end if**
19:     **end for**
20:     **if** $B_{\text{mix}} \ne \emptyset$ **then**
21:        Update $\pi_\theta$ on $B_{\text{mix}}$ using Eq. equation 2.
22:     **end if**
23: **end for**

---

## D VISUALIZATION OF CURRICULUM DYNAMICS

To visually demonstrate that CLPO's online curriculum learning mechanism accelerates the model's learning process more effectively than standard GRPO, we conducted a visual analysis of the problem difficulty distribution within the dynamically constructed training batches. As illustrated in Figure 6, the two methods exhibit starkly different **Learning Dynamics**.

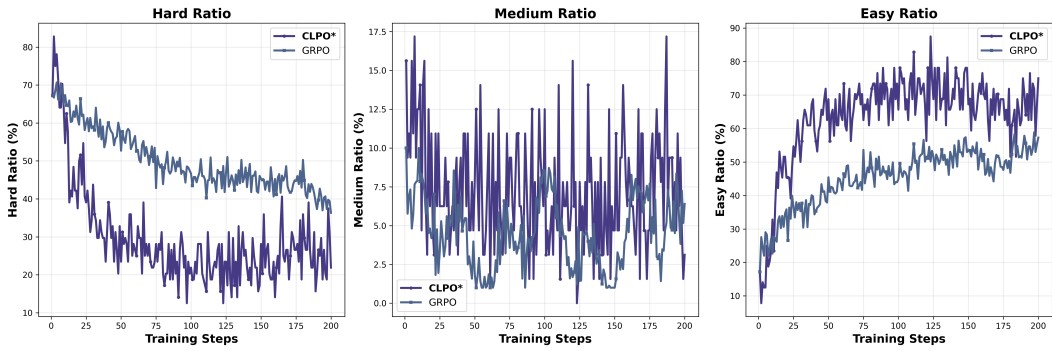

Figure 6: Comparison of the evolution of problem difficulty distribution during training for CLPO (dark purple lines) vs. GRPO (blue-gray lines) over the first 200 training steps. From left to right: Hard Ratio, Medium Ratio, and Easy Ratio.

**Hard Ratio (Left Panel).** The proportion of hard problems for CLPO (dark purple line) shows a **sharp decline** in the early stages of training, quickly converging to a lower level than GRPO. In contrast, the decline for GRPO (blue-gray line) is significantly more gradual, indicating its lower efficiency in tackling difficult problems compared to CLPO's guided approach.

**Medium Ratio (Middle Panel).** The difference in dynamics for medium-difficulty problems is the most revealing, visually demonstrating the different learning modes. CLPO's ratio exhibits strong oscillations around a much higher mean. This serves as direct visual evidence of the dynamic curriculum in action: CLPO continuously simplifies hard problems into learnable medium ones, dynamically replenishing the supply of "nutrients" at the edge of the model's learning zone and thus maintaining a high learning vitality. GRPO's ratio, however, fluctuates within a lower and flatter band, indicating its lack of a mechanism to actively generate medium-difficulty learning opportunities.

**Easy Ratio (Right Panel).** Corresponding to the previous metrics, the proportion of 'Easy/Mastered' problems for CLPO **grows faster and reaches a higher stable level**, directly reflecting its superior overall learning efficiency.

This series of comparisons provides compelling visual evidence that CLPO, through its dynamic curriculum, significantly accelerates the model's learning trajectory and enables a more thorough mastery of the problem space.

## E   PROMPTS FOR ADAPTIVE PROBLEM RESTRUCTURING

This section provides the exact prompts used in our **Adaptive Problem Restructuring (APR)** mechanism to guide the Large Language Model (LLM) in rewriting problems. These prompts are designed to clearly communicate the goal of the restructuring while strictly constraining the model to preserve the mathematical core and the answer of the original problem.

We employ two distinct prompts based on the problem difficulty identified during the Online Curriculum Learning stage. For problems classified as "medium" difficulty, we use the diversification prompt shown in Figure 7, which aims to generate semantically equivalent but stylistically varied versions to enhance the model's generalization. For problems classified as "hard," we use the simplification prompt shown in Figure 8, whose core objective is to reduce the cognitive complexity of the problem and transform it into a more accessible and effective training signal.

## F   CASE STUDIES

To provide a concrete illustration of our Adaptive Problem Restructuring (APR) mechanism in action, this section presents a case study on a medium-difficulty problem. We showcase the original problem and the model's response, followed by the restructured version generated by CLPO and the corresponding improvement in the model's reasoning process.

### F.1   CASE STUDY: DIVERSIFICATION OF A MEDIUM-DIFFICULTY PROBLEM

Figure 9 displays a medium-difficulty problem defined using a compact, symbolic notation typical in mathematical texts. The response generated by the base Qwen3-8B model, while ultimately correct, exhibits a hesitant and circuitous reasoning path. The model frequently expresses self-doubt (e.g., "Wait, but let me check," "Wait a second") and performs multiple, redundant verification steps. This indicates that while the model possesses the necessary knowledge, the symbolic formulation of the problem introduces ambiguity, leading to a less confident and inefficient solution process.

Figure 10 shows the same problem after being processed by CLPO's diversification mechanism. The algorithm restructured the problem by elaborating the compact, symbolic set-builder notation into explicit, full-sentence natural language descriptions. This seemingly minor change had a profound impact on the model's response. The new reasoning path is linear, structured, and confident, proceeding logically from step to step without the self-doubt and redundant checks seen previously.

**Medium Question Rewriting prompt:**

"You are a master-level question rewriting expert. Your mission is TOP-SECRET: rewrite the given math question into a diverse but semantically equivalent version, while maintaining the EXACT same task, constraints, solution method, and CORRECT ANSWER. DO NOT solve the problem or make any assumptions about its solution.\n\n"

"CRITICAL RULES FOR OUTPUT:\n"

"- OUTPUT MUST BE A SINGLE CODE BLOCK using the exact format provided below:\n"

"```text\n"

"user\n"

"Solve the following math problem step by step. The last line of your response should be of the form Answer: $Answer (without quotes) where $Answer is the answer to the problem.\n\n"

"[Rewritten question here]\n\n"

"Remember to put your answer on its own line after \"Answer:\".\n"

"assistant\n"

"```\n"

"- Output NOTHING outside the code block.\n"

"- Rewrite the question to DIVERSIFY its expression but PRESERVE its exact meaning and constraints.\n"

"- Keep the rewritten question STRICTLY UNDER 400 tokens, including all characters, symbols, and spaces.\n"

"- DO NOT change any math content (variables, relationships, solution method, etc.).\n"

"- Avoid repetitive sentence structures or template-like phrasing by using varied grammar structure, synonyms, or logical order.\n"

"- Preserve ALL formatting or directive rules exactly as in the original question (e.g., 'write the answer in a box', 'solve step by step').\n"

"- The rewritten question must lead to the SAME correct answer as the original: {ANSWER}. DO NOT explicitly, implicitly, or indirectly REVEAL or SUGGEST the answer.\n\n"

"HOW TO DIVERSIFY:\n"

"1) Rephrase clauses logically (e.g., change phrasing while maintaining meaning).\n"

"  * Original: 'What is the sum of 4 and 5?' ➡ 'Calculate the result of adding 4 with 5.'\n"

"2) Rearrange sentence structure without altering mathematical intent.\n"

"  * Original: 'Evaluate 3x - 1 where x = 2.' ➡ 'Find the value of 3x minus 1, with x set to 2.'\n"

"3) Use synonyms or alternative phrasing to express the same logic and tasks.\n"

"  * Original: 'Determine the area of the rectangle.' ➡ 'Find the rectangular region's area.'\n"

"4) DO NOT add or remove constraints, context, or instructions.\n\n"

"ONE-SHOT EXAMPLE (STRICT FORMAT ONLY):\n\n"

"Original question (with role markers):\n"

"user\n"

"Compute the value of 2 + 3. The last line of your response should be of the form Answer: $Answer.\n\n"

"Remember to put your answer on its own line after \"Answer:\".\n"

"assistant\n\n"

"Your output:\n"

"```text\n"

"user\n"

"Solve the following math problem step by step. The last line of your response should be of the form Answer: $Answer (without quotes) where $Answer is the answer to the problem.\n\n"

"Determine the result of adding 2 and 3.\n\n"

"Remember to put your answer on its own line after \"Answer:\".\n"

"assistant\n"

"```\n\n"

"ORIGINAL_QUESTION (to rewrite):\n"

"{Q}\n"

Figure 7: The prompt used for diversification restructuring of medium-difficulty problems.

**Hard Question Rewriting prompt:**

"You are a master-level question rewriting expert. Your mission is TOP-SECRET: rewrite the given math question so that it is simpler, clearer, and more direct, while maintaining the EXACT same task, constraints, solution method, and CORRECT ANSWER. DO NOT solve the problem or make any assumptions about its solution.\n\n"

"CRITICAL RULES FOR OUTPUT:\n"

"- OUTPUT MUST BE A SINGLE CODE BLOCK using the exact format provided below:\n"

"```text\n"

"user\n"

"Solve the following math problem step by step. The last line of your response should be of the form Answer: $Answer (without quotes) where $Answer is the answer to the problem.\n\n"

"[Rewritten question here]\n\n"

"Remember to put your answer on its own line after \"Answer:\".\n"

"assistant\n"

"```\n"

"- Output NOTHING outside the code block.\n"

"- Do NOT include any reasoning, parsing, or explanation of the problem (e.g., avoid phrases like \"Let me simplify this question\" or \"This problem can be rewritten as...\").\n"

"- Rewrite the question to make it SIMPLER and CLEARER for the reader, while preserving the EXACT task, math content, and structure.\n"

"- Keep the rewritten question STRICTLY UNDER 400 tokens, including all characters, symbols, and spaces.\n"

"- Remove redundancy and completely clarify definitions, if needed.** For example, if there are implied constraints, make them explicit.\n"

"- Preserve ALL formatting or directive rules exactly as in the original question (e.g., 'write the answer in a box', 'solve step by step').\n"

"- The rewritten question must lead to the SAME correct answer as the original: {ANSWER}. DO NOT explicitly, implicitly, or indirectly REVEAL or SUGGEST the answer.\n\n"

"HOW TO SIMPLIFY:\n"

"1) Eliminate redundant or overly verbose phrasing while keeping the exact math concepts, symbols, and structure.\n"

"2) Avoid complex clauses by breaking them into shorter, simpler sentences, but do NOT remove or alter necessary constraints.\n"

"3) Include IMPLICIT constraints explicitly if they improve clarity (e.g., 'x > 0' if it is implied implicitly).\n"

"4) DO NOT add any background information, new context, or reasoning unrelated to the question.\n\n"

"ONE-SHOT EXAMPLE (STRICT FORMAT ONLY):\n\n"

"Original question (with role markers):\n"

"user\n"

"Compute the value of 2 + 3. The last line of your response should be of the form Answer: $Answer.\n\n"

"Remember to put your answer on its own line after \"Answer:\".\n"

"assistant\n\n"

"Your output:\n"

"```text\n"

"user\n"

"Solve the following math problem step by step. The last line of your response should be of the form Answer: $Answer (without quotes) where $Answer is the answer to the problem.\n\n"

"What is the sum of 2 and 3?\n\n"

"Remember to put your answer on its own line after \"Answer:\".\n"

"assistant\n"

"```\n\n"

"ORIGINAL_QUESTION (to rewrite):\n"

"{Q}\n"

Figure 8: The prompt used for simplification restructuring of hard-difficulty problems.

**ORIGINAL QUESTION**

Solve the following math problem step by step. The last line of your response should be of the form Answer: $Answer (without quotes) where$ Answer is the answer to the problem.

Let set $A = \left\{ n \mid \frac{n}{3} \in \mathbf{N}_+ \right\}$, and $B = \left\{ y \mid y = x + 4 + \sqrt{5 - x^2} \right\}$. Find the number of elements in the set $A \cap B$.

Remember to put your answer on its own line after "Answer:".

**ORIGINAL RESPONSE**

Okay, let's try to solve this problem step by step. First, let me make sure I understand the question correctly. We have two sets, $A$ and $B$. Set $A$ is …So, $A$ is $\{3, 6, 9, 12, \dots\}$ and so on. Set $B$ is defined as …So, $y = x + 4 + \sqrt{5 - x^2}$. The problem is asking for the number of elements in the intersection of $A$ and $B$, meaning the numbers that are in both sets.

So, first, I need to figure out what elements are in set $B$. Then, find the common elements between $A$ and $B$. Let me start by understanding set $B$ more deeply.

…

Solving this inequality:

$$5 - x^2 \geq 0 \Rightarrow x^2 \leq 5 \Rightarrow -\sqrt{5} \leq x \leq \sqrt{5}$$

So, $x$ has to be between $-\sqrt{5}$ and $\sqrt{5}$.

…

Let me try to find the minimum and maximum values of $y(x)$ over this interval. Since $x$ and $\sqrt{5 - x^2}$ are both functions of $x$, maybe I can find their maximum and minimum values.

…

Therefore, the maximum value of $y$ is at $x = \sqrt{5/2} \approx 1.5811$, which is approximately 7.1623, and the minimum value of $y$ occurs at the endpoints.

**Wait**, but when $x = -\sqrt{5}$, $y \approx 1.764$, which is lower than the other endpoint $x = \sqrt{5}$, which is $\approx 6.236$. So the minimum value of $y$ is approximately 1.764, and maximum is approximately 7.1623. Therefore, the range of set $B$ is $y \in [\approx 1.764, \approx 7.1623]$.

…

Therefore, the possible elements of $A$ that can be in $B$ are $3, 6$. Because 9 is already larger than 7.16, so it cannot be achieved by any $y$.

**Wait, but let me check.** Let me verify if 3 and 6 are indeed attainable

…

Therefore, only 3 and 6. But also, what about other numbers?

…

Therefore, the number of elements is 2.

**Wait, but wait! Wait a second**. I need to check if there are other values. Let me check if maybe $y$ can take $3, 6$, or perhaps another multiple of 3?

…

Thus, both 3 and 6 are indeed in set $B$.

### Final Answer

Answer:  2

Figure 9: The original medium-difficulty problem presented in a symbolic style and the corresponding hesitant response generated by the Qwen3-8B model.

This case study vividly demonstrates the effectiveness of our APR mechanism. By transforming a problem into a linguistic format that is more aligned with the model's pre-training data, CLPO is able to reduce ambiguity and significantly improve the robustness and efficiency of the reasoning process itself, not just the final answer.

---

**REWRITTEN QUESTION**

Solve the following math problem step by step. The last line of your response should be of the form Answer: $Answer$ (without quotes) where Answer is the answer to the problem.

Let $A$ be the set of all positive integers $n$ such that $\frac{n}{3}$ is a positive integer. Define $B$ as the set of all real numbers $y$ that can be expressed as $y = x + 4 + \sqrt{5 - x^2}$ for some real number $x$. Determine the number of elements in the intersection of sets $A$ and $B$.

Remember to put your answer on its own line after "Answer:".

**REWRITTEN RESPONSE**

Okay, let's try to tackle this problem step by step. First, I need to understand what sets $A$ and $B$ are and then find their intersection. Starting with set $A$: it's defined as …. So set $A$ is all positive multiples of 3. **Got that.** Now, set $B$ is a bit trickier. It's the set of …

First, I need to figure out the domain of $x$ for which $\sqrt{5 - x^2}$ is real.

…

Therefore, $x$ must be in the interval $[-\sqrt{5}, \sqrt{5}]$. So $x$ is between approximately $-2.236$ and $2.236$.

…

But maybe **it's better to first find** the range of possible $y$'s in set $B$, and then check which multiples of 3 lie within that range. Let's try that approach.

So, first, find the minimum and maximum possible values of $y$.

…

Therefore, set $B$ contains all real numbers $y$ between $4 - \sqrt{5}$ and $4 + \sqrt{10}$. Now, set $A$ is the set of positive integers that are multiples of 3. So $A = \{3, 6, 9, 12, \dots\}$ and so on. Now, to find the intersection of $A$ and $B$, **we need to find** all positive integers $n$ that are multiples of 3 and lie between $4 - \sqrt{5}$ and $4 + \sqrt{10}$.

…

So in that interval, only 3 and 6. Hence, the intersection of $A$ and $B$ has two elements. So, the answer is 2. Therefore, the number of elements in the intersection is 2.

### Final Answer

Answer: 2

---

Figure 10: The restructured version of the problem, elaborated into natural language by CLPO, and the resulting confident, linear reasoning path from the same model.

