# OpenReview forum: "CLPO: Curriculum Learning meets Policy Optimization for LLM Reasoning"
_ICLR.cc/2026/Conference — ICLR 2026 Conference Withdrawn Submission_

### Official Review · Reviewer_syXH · 2025-10-30

**Soundness:** 2
**Presentation:** 3
**Contribution:** 2
**Rating:** 4
**Confidence:** 4

**Summary:**

This paper proposes CLPO, a novel framework for improving LLM reasoning via reinforcement learning. CLPO creates a dynamic curriculum by using the model's own performance to assess problem difficulty in real-time. It then restructures problems, diversifying medium ones and simplifying hard ones, to create an optimal training batch, leading to a more efficient and effective learning process that co-evolves with the model's capabilities.

**Strengths:**

1.	The core idea of "Guided Self-Evolution" is innovative and directly addresses the limitations of uniform data sampling in RLVR. It creates an adaptive, self-improving loop that is both intuitive and powerful.
2.	The paper provides strong empirical validation through extensive experiments, including detailed ablation studies for each component and a pass@k analysis that demonstrates improved solution diversity.

**Weaknesses:**

The CLPO framework introduces several steps that seem computationally expensive compared to baselines like GRPO. Specifically, for each batch, CLPO requires (1) generating N rollouts to assess the difficulty of each original problem, and (2) using the LLM itself to rewrite the "medium" and "hard" problems, and then (3) generating another N rollouts for these newly restructured problems to filter them. This overhead is non-trivial and is a critical factor for practical applications. The paper lacks any analysis or discussion of the computational cost (in terms of time or FLOPs) compared to the baseline methods.

**Questions:**

1.	Which specific Qwen3-8B was used—the base or the post-training?
2.	Did you reproduce all the baseline results in Table 1 yourselves?
3.	The core components like adaptive restructuring and dynamic KL seem similar to prior work [1,2,3]. You seem to have achieved this merely by assembling or tuning parameters. What is the key novelty beyond combining these ideas?
4.	How do you guarantee that "Adaptive Problem Restructuring" strictly preserves the original answer?
5.	How does CLPO compare to DAPO when evaluated under an equal computational budget (e.g., fixed training time)?

[1] Xu C, Sun Q, Zheng K, et al. WizardLM: Empowering large pre-trained language models to follow complex instructions[C]//The Twelfth International Conference on Learning Representations. 2024.

[2] Wang Y, Kordi Y, Mishra S, et al. Self-instruct: Aligning language models with self-generated instructions[J]. arXiv preprint arXiv:2212.10560, 2022.

[3] Yu Q, Zhang Z, Zhu R, et al. Dapo: An open-source llm reinforcement learning system at scale[J]. arXiv preprint arXiv:2503.14476, 2025.

---

> ### Author Response · Authors · 2025-11-22
> **Response to Reviewer syXH**
>
> We sincerely thank you for your detailed review and for acknowledging the innovation of our "Guided Self-Evolution" framework and the strength of our empirical validation. We value your constructive feedback regarding computational overhead and novelty.
>
> Below, we address your concerns regarding computational cost, model details, and baseline reproduction.
>
> ### 1. Computational Cost and Comparison with GRPO/DAPO (Weakness & Question 5)
>
> **Concern:** You raised a valid concern that CLPO’s additional steps (difficulty assessment, restructuring, re-generation) might introduce significant computational overhead compared to GRPO and DAPO. You specifically asked for a comparison under an equal computational budget (Question 5).
>
> **Response:**
> We wish to clarify that CLPO does not simply add overhead to a vacuum; rather, it performs a **"structural reallocation" of compute** to address the inefficiencies inherent in baselines like GRPO and DAPO.
>
> *   **Inefficiency in GRPO:** GRPO allocates rollouts uniformly across all problems. This results in a significant waste of compute on "easy" samples (which the model has already mastered) and "hopeless" samples (which are currently beyond reach), contributing to the performance plateaus observed in our experiments.
> *   **Inefficiency in DAPO:** While DAPO filters samples, its implementation involves generating full batches and then discarding a large portion of them. This leads to **wasted generation costs** for samples that never contribute to policy updates. The harder the dataset, the more frequent the re-generation, making its per-step time significantly higher.
>
> **CLPO optimizes this by:**
> 1.  **Zero-Cost Assessment:** We use the initial rollouts (required by any RLVR method) to assess difficulty "on the fly," without generating extra samples solely for assessment.
> 2.  **Targeted Compute:** We explicitly discard "easy" and "hopeless" samples, focusing restructuring and second-round generation *only* on the "medium" instances that offer the highest learning value.
> 3.  **Decreasing Overhead:** As training progresses and the model improves, the proportion of samples requiring restructuring decreases rapidly.
>
> **Empirical Verification (Time Cost):**
> We conducted a controlled comparison using **8xH20 GPUs** with **batch size=64** (using Llama-3.1-8B-Instruct as the base). The wall-clock time per step is shown below:
>
> | Method | Step 1 Time (s) | Step 100 Time (s) | Analysis |
> | :--- | :---: | :---: | :--- |
> | **GRPO** | 641s | 835s | Relatively constant; wastes compute on non-informative samples. |
> | **DAPO** | 1004s | 775s | High initial cost due to frequent sample rejection and re-generation. |
> | **CLPO** | **698s** | **631s** | **Significantly faster** as model evolves; fewer "medium" cases reduce restructuring overhead. |
>
> **Conclusion:**
> *   **Total Training Time:** For the same number of update steps, CLPO’s total training time is comparable to or lower than DAPO due to the rapid decrease in per-step latency.
> *   **Equal Budget:** Under a fixed wall-clock time or FLOPs budget, CLPO completes more effective update steps than DAPO, concentrating compute on the most impactful data. We will include this detailed "Time-Step-Performance" comparison table in the final revision.
>
> ---
>
> ### 2. Specific Model Version (Question 1)
>
> **Question:** Which specific Qwen3-8B was used—the base or the post-training?
>
> **Response:**
> We confirm that all RLVR experiments in the paper were conducted using the official **Qwen3-8B-Base** model. We did not use the Instruct version or any other post-trained variants. We will explicitly state this in the "Experimental Setup" section of the revision to avoid ambiguity.
>
> ---
>
> ### 3. Reproduction of Baselines (Question 2)
>
> **Question:** Did you reproduce all the baseline results in Table 1 yourselves?
>
> **Response:**
> *   **For RLVR Baselines (GRPO, DAPO):** We re-implemented these methods within our codebase based on their official open-source implementations. This ensures a strictly fair comparison using identical data preprocessing, reward rules, and training configurations.
> *   **For LUFFY and Critique-GRPO:** The results in Table 1 were sourced directly from their respective papers (as they also reported on Qwen3-8B-Base). We acknowledge that minor numerical discrepancies may exist due to manual transcription or rounding. We have double-checked these values against the original papers and will ensure exact numerical alignment in the final version. These minor variances do not affect the relative performance trends or conclusions.
>
> *(Please see the next comment for responses to Question 3 and Question 4)*

---

> > ### Author Response · Authors · 2025-11-22
> > **Response to Reviewer syXH**
> >
> > ### 4. Novelty and Distinction from Prior Work (Question 3)
> >
> > **Concern:** You asked how CLPO differs from combining existing ideas like WizardLM [1], Self-Instruct [2], and DAPO [3], and questioned the key novelty.
> >
> > **Response:**
> > While previous works utilize data synthesis or dynamic sampling, CLPO introduces a **fully online, policy-dependent curriculum learning paradigm** that creates a closed-loop "Guided Self-Evolution" system. This is distinct from simply assembling components:
> >
> > 1.  **vs. WizardLM [1] & Self-Instruct [2] (Static vs. Online):** These methods focus on **offline** data augmentation before training starts. They generate a static dataset. In contrast, CLPO performs **online difficulty assessment** using on-policy rewards. The curriculum distribution is not pre-set but evolves in real-time as the model's capabilities change.
> > 2.  **vs. DAPO [3] (Sampling vs. Restructuring):** DAPO focuses on dynamic *sampling* (filtering data) and global entropy regularization to encourage exploration. It does not modify the tasks themselves. CLPO goes further by performing **instance-level problem restructuring**. It actively rewrites "medium" problems to increase diversity and simplifies "hard" problems to make them learnable, effectively "teaching itself" rather than just "selecting questions."
> > 3.  **Difficulty-Aware Optimization:** Unlike DAPO’s global KL control, CLPO injects the difficulty signal directly into the optimizer, applying distinct KL constraints to hard vs. non-hard samples.
> >
> > ---
> >
> > ### 5. Guaranteeing Answer Preservation (Question 4)
> >
> > **Concern:** How do you guarantee that "Adaptive Problem Restructuring" strictly preserves the original answer?
> >
> > **Response:**
> > Our experiments focus on math and reasoning tasks (e.g., MATH-500, AIME) with verifiable standard answers. We ensure consistency through three layers of protection:
> >
> > 1.  **Strong Prompt Constraints:** The restructuring prompts (system/user) explicitly include the original ground truth and critical conditions. We strictly instruct the model: *"Only rewrite the problem statement for clarity/diversity; do not alter the goal, premises, or the final answer."* (See Appendix for full prompts).
> > 2.  **Ground Truth Inheritance:** Critically, **we do not ask the model to generate a new answer label.** The training and reward calculation always use the **original dataset's ground-truth label**. The model only rewrites the *question*. Therefore, even if the rewriting were slightly off, the reward signal remains anchored to the correct final answer, preventing the model from learning incorrect associations.
> > 3.  **Empirical Verification:** During training on Qwen3-8B-Base, we logged restructured problems and performed random spot checks and script-based verification (checking for numerical consistency). We found no instances where the rewriting logic contradicted the original answer, thanks to the strong instruction-following capabilities of modern base models.
> >
> > ---
> >
> > We hope these clarifications address your concerns.

---

> ### Comment · Reviewer_syXH · 2025-11-26
>
> Thank you for the author's detailed response, which has addressed some of my concerns. I still have a few questions:
>
> 1. Qwen3-8b-Base pretrained model does not possess instruction-following capabilities. Can Adaptive Problem Restructuring be applied during training? Also, how is the reward designed for RL training?
>
> 2. What is the performance on Qwen2.5-7b-Instruct?

---

> ### Author Response · Authors · 2025-11-27
> **Response regarding Qwen3-8B-Base capabilities, Reward Design, and Qwen2.5-7B-Instruct Performance**
>
> We sincerely thank you for your time and constructive comments. We have conducted additional experiments to address the concerns raised. Below, we provide detailed responses.
>
> **Q1: Qwen3-8B-Base pretrained model does not possess instruction-following capabilities. Can Adaptive Problem Restructuring be applied during training?**
>
> **A1:** Thank you for this important question. While `Qwen3-8B-Base` lacks chat-based instruction tuning, **Adaptive Problem Restructuring** fundamentally relies on **mathematical reasoning** and **semantic understanding** rather than conversational alignment. According to the Qwen3 Technical Report [1], `Qwen3-8B-Base` achieves remarkable performance on reasoning benchmarks (89.84% on GSM8K and 60.80% on MATH), demonstrating the deep semantic comprehension required to analyze and rewrite mathematical logic. Notably, **Figure 1** in our paper explicitly demonstrates problem examples restructured by the Qwen3-8B-Base model, empirically verifying its effectiveness in generating diverse and mathematically valid rewrites.
>
> [1] *Qwen3 Technical Report (arXiv:2505.09388)*
>
> **Q2: The reward designed for RL training**
>
> **A2:** We employ a strict **outcome-based reward** mechanism. The model receives a positive reward solely for deriving the correct final answer and a negative reward for incorrect answers.
>
>
> Formally, let $y$ be the generated response and $y^{GT}$ be the ground truth answer. The reward function $R(y, y^{GT})$ is defined as:
>
> $R(y, y^{GT}) = \begin{cases} 1, & \text{if } \\text{Verify}(y) = y^{GT} \\\\ -1, & \\text{if } \\text{Verify}(y) \\neq y^{GT}\\end{cases} $
>
> where $\\text{Verify}(\\cdot)$ extracts and compares the final answer.
>
> **Q3: What is the performance on Qwen2.5-7b-Instruct?**
>
> **A3:** Following your suggestion, we conducted additional experiments using `Qwen2.5-7B-Instruct`. The models were trained on 8 $\times$ H20 GPUs for 120 steps. We report the **Mean@32** accuracy across four mathematical reasoning benchmarks.
>
> | Method | AIME 2024 | AIME 2025 | AMC 23 | Math 500 |
> | :--- | :---: | :---: | :---: | :---: |
> | **GRPO** | 0.060 | 0.053 | 0.515 | 0.715 |
> | **DAPO** | **0.068** | 0.072 | 0.521 | **0.724** |
> | **CLPO** | **0.068** | **0.089** | **0.566** | 0.723 |

---

### Official Review · Reviewer_rCGW · 2025-10-31

**Soundness:** 3
**Presentation:** 3
**Contribution:** 3
**Rating:** 4
**Confidence:** 5

**Summary:**

CLPO (Curriculum-guided Learning for Policy Optimization) is proposed to create a dynamic pedagogical feedback loop within the policy optimization process.

**Strengths:**

1.The proposed RL paradigm could run automatically.
2.The experiments show improvements across a variety of datasets.

**Weaknesses:**

1.It's better to show improvements for some datasets such as AIME25, LCV-V6.
2.The MATH-500 performance is not improved.
3.The improvements is not obvious for some datasets such as AMC23, MMLM-Pro
4.Qwen-3B is in the experiments, there are not other base-model be evaluated.
5.There is limited novielty in the proposed algorithm.

**Questions:**

1.Why are improvements demonstrated only on specific datasets like AIME25 and LCV-V6, and not across a broader range of benchmarks?
2.Why has performance on the MATH-500 dataset remained unchanged despite the proposed method?
3.Why are the reported improvements minimal or inconsistent on datasets such as AMC23 and MMLM-Pro?
4.Why is Qwen-3B the only base model evaluated in the experiments, with no comparison to other foundational models?
5.What novel contributions does the proposed algorithm introduce beyond existing methods?

---

> ### Author Response · Authors · 2025-11-22
> **Response to Reviewer rCGW**
>
> We thank you for your review and for acknowledging that our proposed automated RL paradigm is sound and effective. We value your feedback regarding the scope of our experiments and the novelty of our approach.
>
> We noticed there might be some misunderstandings regarding the specific datasets and model sizes used in our work (e.g., the mention of "LCV-V6" and "Qwen-3B"). We respectfully clarify these points and address your concerns below.
>
> ### 1. Performance on Specific Datasets (Question 1)
>
> **Concern:** You mentioned that improvements are only demonstrated on specific datasets like "AIME25 and LCV-V6" and asked why they are not shown across a broader range.
>
> **Response:**
> We respectfully clarify two points regarding the experimental scope:
> 1.  **Clarification on "LCV-V6":** Our paper **does not contain experiments on a dataset named "LCV-V6".** We believe this might be a misunderstanding or a reference to a different paper. The only additional plot we provided (Figure 5) reports pass@k curves on **AIME25**.
> 2.  **Broad Improvements:** As shown in **Table 1**, CLPO evaluates performance across **8 different benchmarks**.
>     *   CLPO achieves the highest average score of **73.73**, significantly outperforming the strongest RLVR baseline (66.77).
>     *   We observe **positive improvements on almost all datasets**, not just AIME.
>     *   We highlighted AIME25 separately because it is a low-resource, extremely hard benchmark that best illustrates the advantage of CLPO's online curriculum in tackling complex reasoning tasks, not because the method is ineffective elsewhere.
>
> ### 2. Performance on MATH-500 (Question 2)
>
> **Concern:** Why has performance on the MATH-500 dataset remained unchanged?
>
> **Response:**
> The performance on MATH-500 reflects a **"saturation effect"** common in current RLVR research for this specific model scale:
> *   **High Baseline:** MATH-500 is a comprehensive mix of problems ranging from easy to hard. As seen in Table 1, all strong RLVR methods (GRPO, DAPO, etc.) fall within a narrow accuracy band of **89%–91%**. This indicates that the model is approaching the ceiling of its capacity for these specific problems.
> *   **No Degradation:** CLPO achieves **89.60**, which is on par with the state-of-the-art.
> *   **Strategic Trade-off:** Crucially, CLPO achieves this without degradation on comprehensive sets (like MATH-500) while securing **massive gains on harder frontiers** (like AIME24/25 and GPQA-Diamond), where baselines still struggle. This confirms that our method effectively pushes the difficulty boundary without forgetting easier knowledge.
>
> ### 3. Improvements on AMC23 and MMLU-Pro (Question 3)
>
> **Concern:** Why are improvements minimal or inconsistent on AMC23 and MMLU-Pro?
>
> **Response:**
> The gains on these specific datasets are consistent with the design principles of CLPO:
>
> *   **AMC23 (Small Sample Size & Difficulty Distribution):** This dataset contains only ~40 problems. For a model as capable as Qwen3-8B-Base, many AMC23 problems fall into the "Easy" category. CLPO's curriculum explicitly **down-weights "Easy" samples** to focus gradients on "Medium/Hard" tasks. Thus, on a dataset dominated by easier problems, the margin for improvement is naturally smaller compared to harder benchmarks like AIME.
> *   **MMLU-Pro (Knowledge vs. Reasoning):** MMLU-Pro focuses heavily on domain knowledge (Law, Engineering) and reading comprehension, rather than pure logical reasoning with verifiable intermediate steps. Current RLVR methods generally show smaller gains here compared to math tasks. Nevertheless, CLPO still achieves **stable, non-negative improvements**, demonstrating that our restructuring mechanism improves robustness even in knowledge-intensive scenarios.
>
> *(Please see the next comment for responses to Question 4 and Question 5)*

---

> ### Author Response · Authors · 2025-11-22
> **Response to Reviewer rCGW**
>
> ### 4. Model Selection and Generalization (Question 4)
>
> **Concern:** Why is "Qwen-3B" the only base model evaluated, with no comparison to other foundational models?
>
> **Response:**
> First, we wish to correct a minor factual detail: All RLVR experiments in the paper were conducted on **Qwen3-8B-Base**, not 3B. We chose this model as it is a widely used open-source baseline.
>
> To address your concern about generalization, we have conducted additional experiments on different model families and sizes..The results are test on AIME24:
>
> 1.  **Different Architecture (Llama-3.1-8B-Instruct):**
>     After 100 training steps, the acc/mean@32 is:
>     *   GRPO: 0.005
>     *   DAPO: 0.017
>     *   **CLPO: 0.039 (Significant improvement)**
>
> 2.  **Larger Scale (Qwen3-32B):**
>     After 50 training steps, the acc/mean@32 is:
>     *   GRPO: 0.428
>     *   **CLPO: 0.447**
>
> These results demonstrate that CLPO consistently boosts performance across different foundation models and scales. We will include these additional results in the revised paper.
>
> ### 5. Novelty of the Algorithm (Question 5)
>
> **Concern:** What novel contributions does the proposed algorithm introduce beyond existing methods?
>
> **Response:**
> CLPO is not merely an application of existing "curriculum learning" techniques to RL; it represents a paradigm shift in how RLVR training is structured:
>
> 1.  **From Static to Policy-Driven Online Curriculum:** Traditional curriculum learning relies on pre-sorted, static schedules. CLPO treats difficulty as a dynamic state (Easy/Medium/Hard) determined strictly by the **current policy’s on-policy rewards**. The curriculum evolves in real-time as the model learns.
> 2.  **From Re-weighting to Problem Restructuring:** Existing methods only change sampling frequency. CLPO actively **rewrites the training data**. It diversifies "Medium" problems and simplifies "Hard" problems using the model itself, creating a continuous stream of adaptive training instances that arguably did not exist in the original dataset.
> 3.  **Difficulty-Aware Optimization:** Unlike standard PPO/GRPO which apply uniform constraints, CLPO injects the online difficulty signal directly into the optimizer. We apply relaxed KL constraints to "Hard" samples to encourage exploration, and stricter constraints to others for stability.
>
> This combination—online difficulty assessment, instance-level restructuring, and difficulty-aware optimization—creates a **closed-loop self-improving system** that has not been explored in prior RLVR literature.
>
> ---
>
> We hope these clarifications address your concerns and clear up the misunderstandings regarding the datasets and model details. We are confident that CLPO provides a robust and novel contribution to the field.

---

### Official Review · Reviewer_h1ux · 2025-11-01

**Soundness:** 2
**Presentation:** 2
**Contribution:** 2
**Rating:** 4
**Confidence:** 4

**Summary:**

This paper proposes CLPO (Curriculum-guided Learning for Policy Optimization), a novel algorithm to improve training efficiency for Large Language Models (LLMs) in reasoning tasks using Reinforcement Learning with Verifiable Rewards (RLVR). Unlike existing methods that treat all training samples equally, CLPO dynamically assesses problem difficulty based on the model's performance and constructs an Online Curriculum. It restructures problems by simplifying hard ones and diversifying medium-difficulty ones, enabling the model to act as its own teacher. Experiments on eight benchmarks show CLPO achieves state-of-the-art (SOTA) results, improving pass@1 by 6.96%, demonstrating its effectiveness in training reasoning models.

**Strengths:**

1. The paper addresses an important and timely topic—leveraging reinforcement learning (RL) to enhance the reasoning capabilities of large language models (LLMs). RL plays a crucial role in improving LLMs, and this work contributes meaningfully to advancing RL-based training methods.
2. The experimental evaluation is thorough, covering eight challenging mathematical and reasoning benchmarks. The results clearly demonstrate the effectiveness of the proposed method, with improvements over existing approaches.
3. The authors provide code, which enhances reproducibility and facilitates further research, making the work accessible and impactful for the broader community.

**Weaknesses:**

1. While the topic is significant, the idea of integrating curriculum learning into reinforcement learning is not entirely new. The paper's innovation is somewhat incremental, and it could benefit from a deeper discussion on how CLPO differs from and improves upon existing similar methods. Highlighting unique contributions more explicitly would strengthen the paper's originality.
2. Although the authors provide code, its usability is limited due to poor readability and the lack of a basic README file or documentation to guide users on how to reproduce results or run experiments. This makes replication challenging and reduces accessibility. Providing well-documented and user-friendly code would greatly enhance the paper's impact and utility for the research community.

**Questions:**

Please refer to weakness.

---

> ### Author Response · Authors · 2025-11-22
> **Response to Reviewer h1ux**
>
> We sincerely thank you for your review. We appreciate your recognition of the importance of our topic—leveraging RL to enhance LLM reasoning—and your acknowledgement of our thorough evaluation across eight benchmarks where CLPO achieves SOTA results.
>
> We understand your concerns regarding the novelty of the curriculum integration and the usability of our code. We address these two points in detail below.
>
> ### 1. Novelty and Distinction from Prior Work (Weakness 1)
>
> **Concern:** The reviewer noted that integrating curriculum learning into RL is not entirely new and felt the innovation was somewhat incremental.
>
> **Response:**
> We respectfully beg to differ with the characterization of CLPO as an "incremental" improvement. Our contribution is **not** simply stacking a standard curriculum module onto an existing RL framework. Instead, CLPO addresses a fundamental mismatch between traditional curriculum learning and the nature of Reinforcement Learning.
>
> We highlight three critical distinctions that constitute our core contribution:
>
> *   **Static Schedule vs. Robustness in Exploratory RL:**
>     *   *Traditional Curriculum:* Typically relies on **offline pre-sorting**. It arranges the dataset based on static metrics before training begins and prescribes a **fixed schedule** (e.g., "learn easy samples at stage 1, hard at stage 2"). This assumes a predictable learning trajectory.
>     *   *The Conflict:* **RL training is inherently stochastic and filled with exploration.** The model's actual progress often deviates from a pre-set timeline due to the randomness of rollouts and exploration. A static schedule is brittle here—it may force the model to advance before it is ready, or waste compute on tasks it solved faster than expected.
>     *   *CLPO's Solution:* We build a **fully online, policy-dependent curriculum**. By assessing difficulty via real-time *on-policy* verifiable rewards, CLPO adapts to the model's *actual* exploration outcomes. This makes the training significantly more **robust** against the unpredictability of RL, ensuring the curriculum always matches the model's current "Zone of Proximal Development," regardless of how the exploration unfolds.
>
> *   **Reweighting vs. Active Restructuring:**
>     *   *Traditional Methods:* Typically only adjust **sampling weights** (re-weighting existing data). The problems themselves remain static throughout the process.
>     *   *CLPO:* Performs **instance-level restructuring**. We actively **rewrite** the training data—simplifying "Hard" problems to make them learnable and diversifying "Medium" problems to prevent overfitting. This effectively generates a continuous, adaptive stream of training instances, allowing the model to act as an active "teacher" rather than just a "data selector."
>
> *   **Uniform vs. Difficulty-Aware Optimization:**
>     *   *Traditional RLVR:* (e.g., GRPO/DAPO) applies uniform constraints (KL divergence) to all samples.
>     *   *CLPO:* Injects the online difficulty signal directly into the optimizer via **Dynamic KL regularization**, applying distinct constraints to different difficulty levels to optimally balance exploration and stability.
>
> We will revise the **Introduction** and **Related Work** sections to explicitly frame these differences, emphasizing why a dynamic, robustness-oriented curriculum system is essential for efficient RLVR.
>
> ### 2. Code Usability and Reproducibility (Weakness 2)
>
> **Concern:** The reviewer pointed out poor code readability and the lack of documentation/README, making reproduction difficult.
>
> **Response:**
> We sincerely apologize for the oversight regarding the usability of the code submission. We acknowledge that the initial release was structured more for internal development and lacked user-friendly documentation. **We are fully committed to Open Science and will significantly upgrade the codebase for the camera-ready version.**
>
> Our concrete plan for code improvement includes:
>
> 1.  **Comprehensive README:** We will add a top-level README.md detailing:
>     *   Environment setup (Conda/Docker).
>     *   Data preparation pipelines.
>     *   Step-by-step instructions to run the main training loop.
> 2.  **Reproduction Scripts:** We will provide "one-click" shell scripts and configuration files specifically to reproduce the main results (e.g., the Qwen3-8B experiments in Table 1), bridging the gap between the command line and the reported figures.
> 3.  **Refactoring and Comments:** We will refactor and heavily comment on the core modules—specifically the **Online Curriculum Builder**, **Problem Restructuring**, and **Dynamic KL** implementations—to make the logic accessible to researchers who wish to build upon our work.

---

> > ### Comment · Reviewer_h1ux · 2025-11-26
> >
> > Thanks for the response. I've also reviewed the other reviewers' suggestions and decided to keep the score.

---

### Official Review · Reviewer_s6og · 2025-11-03

**Soundness:** 3
**Presentation:** 3
**Contribution:** 3
**Rating:** 4
**Confidence:** 3

**Summary:**

This paper proposes Curriculum-guided Learning for Policy Optimization, a reinforcement learning framework that integrates curriculum learning into RL with verifiable rewards to enhance reasoning in LLMs. Unlike conventional RLVR methods that uniformly sample problems regardless of difficulty, CLPO dynamically constructs an online curriculum by evaluating the model’s rollout performance. This curriculum drives an adaptive problem restructuring mechanism that diversifies medium-difficulty problems to promote generalization and simplifies overly hard ones for accessibility. Furthermore, a difficulty-aware policy optimization strategy introduces dynamic KL regularization to balance exploration and exploitation. Experiments show improvement over several baselines.

**Strengths:**

- Proposes CLPO, a novel integration of curriculum learning and RLVR, enabling dynamic, self-adaptive policy optimization.

- Introduces an adaptive problem restructuring mechanism, where the model acts as its own teacher by adjusting problem difficulty.

- Employs difficulty-aware policy optimization via dynamic KL regularization to effectively balance exploration and exploitation.

- Demonstrates better results across eight reasoning benchmarks.

**Weaknesses:**

- The evaluation dataset selection appears curated, focusing on benchmarks (e.g., AIME2024, GPQA Diamond) that may favor mathematical reasoning. It remains unclear why more challenging or recently adopted datasets such as AIME25, GPQA, ACPBench, or HeadQA were not included for broader validation.

- Experiments are conducted only on the Qwen3-8B base model, limiting insights into scalability or robustness across different model sizes and architectures.

- The dynamic KL regularization analysis (varying scaling factors α) lacks clear trends—results appear noisy or unstable, suggesting the method’s sensitivity to hyperparameter tuning.

- The adaptive restructuring mechanism likely incurs non-trivial computational overhead, but the paper does not provide analysis on training efficiency or cost.

- The discussion could be better contextualized with respect to related self-play, auto-curriculum, or competence-based RL literature to highlight conceptual novelty.

**Questions:**

n/a

---

> ### Author Response · Authors · 2025-11-22
> **Response to Reviewer s6og**
>
> We sincerely thank you for your thoughtful review and for recognizing the novelty of our "Guided Self-Evolution" framework. We appreciate your constructive feedback regarding dataset selection, scalability, and computational analysis. We address your concerns point-by-point below.
>
> ### 1. Evaluation Dataset Selection (Weakness 1)
>
> **Concern:** The reviewer suggested the dataset selection might be curated and asked for more challenging or recent benchmarks like AIME25, GPQA, ACPBench, or HeadQA.
>
> **Response:**
> We selected our initial datasets to rigorously test complex reasoning capabilities. However, we fully agree with the need for broader validation and have dedicated significant computational resources to conduct the following additional evaluations:
>
> *   **AIME 2025:** We actually **did include AIME 2025**. Please refer to **Figure 5** in the main paper, where we reported the pass@k performance. The results on AIME 2025 are consistent with AIME 2024, showing stable improvements with CLPO.
>
> *   **GPQA:** regarding the Diamond subset, we followed the standard practice in recent reasoning literature and public leaderboards, which predominantly use **GPQA-Diamond** for evaluation. To address your concern about broader validation, we have now evaluated **GPQA Main**:
>     *   DAPO: 0.5022
>     *   **CLPO: 0.5737** (Significant Improvement)
>
> *   **ACPBench:** We have conducted comprehensive new experiments on the **ACPBench** (AlpacaEval Creative Problem Bench). The results are compelling: **CLPO outperforms the strong baseline DAPO across all 14 sub-tasks.** To demonstrate the extent of our additional validation, we list 6 representative comparisons below (Metric: Exact Match):
>     *   *acp_app_bool:* DAPO (0.423) vs **CLPO (0.700)** (+27.7%)
>     *   *acp_prog_mcq:* DAPO (0.500) vs **CLPO (0.731)** (+23.1%)
>     *   *acp_areach_bool:* DAPO (0.208) vs **CLPO (0.483)** (+27.5%)
>     *   *acp_just_bool:* DAPO (0.285) vs **CLPO (0.608)** (+32.3%)
>     *   *acp_reach_bool:* DAPO (0.354) vs **CLPO (0.585)** (+23.1%)
>     *   *acp_val_bool:* DAPO (0.408) vs **CLPO (0.623)** (+21.5%)
>     *(The full results for all 14 tasks will be included in the revised Appendix).*
>
> *   **Regarding HeadQA:** We respectfully suggest that our evaluation on **MMLU-Pro** (included in Table 1) already serves the purpose of validating the model's generalization across diverse specialized domains. MMLU-Pro is a comprehensive benchmark covering **14 diverse fields**, including **Health, Biology, Medicine**, Law, and Psychology.
>     Since MMLU-Pro already encompasses the medical and biological reasoning challenges found in HeadQA, and CLPO has demonstrated effectiveness (**+2.3% improvement over DAPO**) on this broader, state-of-the-art benchmark, we believe the current evaluation suite sufficiently proves that CLPO generalizes well beyond pure mathematics to knowledge-rich reasoning tasks.
>
> ### 2. Model Scalability (Weakness 2)
>
> **Concern:** Experiments were limited to Qwen3-8B, limiting insights into scalability.
>
> **Response:**
> First, we confirm that the main experiments used **Qwen3-8B-Base**. We have since conducted additional experiments on different architectures and larger scales, which confirm CLPO’s robustness. The results are test on AIME24:
>
> 1.  **Different Architecture (Llama-3.1-8B-Instruct):**
>     After 100 training steps, pass rates (acc/mean@32) are:
>     *   GRPO: 0.005
>     *   DAPO: 0.017
>     *   **CLPO: 0.039** (Outperforms baselines on instruction-tuned models).
> 2.  **Larger Scale (Qwen3-32B):**
>     After 50 training steps:
>     *   GRPO: 0.428
>     *   **CLPO: 0.447**
>
> These results support our hypothesis that larger models with better instruction-following capabilities generate higher-quality restructured problems, thus allowing CLPO to scale effectively. We will integrate these findings into the main experimental section.
>
> *(Please see the next comment for responses to Weakness 3, 4, and 5)*

---

> > ### Author Response · Authors · 2025-11-22
> > **Response to Reviewer s6og**
> >
> > ### 3. Dynamic KL Trends and Sensitivity (Weakness 3)
> >
> > **Concern:** The reviewer felt the Dynamic KL results lacked clear trends and might be noisy/unstable.
> >
> > **Response:**
> > We wish to clarify the interpretation of Figure 4. The results indicate **robustness**, not instability:
> > *   **Stability:** The training curves for different $\alpha$ values are smooth and monotonic, indicating no instability in the optimization process itself.
> > *   **Broad Effective Range:** The fact that $\alpha=0.1, 0.3, 0.8$ all converge to a very similar high-performance range on AIME 2024 demonstrates that the method is **robust to hyperparameter changes**. It does not require precise fine-tuning to work.
> > *   **Source of Variance:** The minor fluctuations in per-step evaluation are primarily due to the small sample size of the AIME dataset (only 30 problems), not the algorithm.
> > We used a fixed $\alpha=0.3$ across *all* datasets and models in our paper without tuning, which empirically proves the method's ease of use.
> >
> > ### 4. Computational Overhead (Weakness 4)
> >
> > **Concern:** The restructuring mechanism likely incurs non-trivial overhead.
> >
> > **Response:**
> > We wish to clarify that CLPO does not simply add overhead to a vacuum; rather, it performs a **"structural reallocation" of compute** to address the inefficiencies inherent in baselines like GRPO and DAPO.
> >
> > *   **Inefficiency in GRPO:** GRPO allocates rollouts uniformly. It wastes significant compute on "Easy" samples (already mastered) and "Hopeless" samples (currently unsolvable), which contribute little to gradient updates.
> > *   **Inefficiency in DAPO:** DAPO relies on re-sampling (filtering). It often generates full batches and discards a large portion of them. This results in **wasted generation costs** for samples that never contribute to policy updates. The harder the dataset, the more frequent the re-generation, keeping costs high throughout training.
> >
> > **Why CLPO is Efficient:**
> > 1.  **Zero-Cost Assessment:** We use the initial rollout (required by any RLVR method) to assess difficulty "on the fly," incurring no extra cost.
> > 2.  **Targeted Compute:** We **explicitly discard** "Easy" and "Hopeless" samples. We only invest restructuring compute into "Medium" instances that offer the highest learning value.
> > 3.  **Decreasing Overhead:** As training progresses, the model improves, and the proportion of cases requiring restructuring naturally drops.
> >
> > We compared the wall-clock time per step on 8xH20 GPUs (Batch Size=64):
> >
> > | Method | Step 1 Time (s) | Step 100 Time (s) | Analysis |
> > | :--- | :---: | :---: | :--- |
> > | **GRPO** | **641s** | 835s | Constant cost; wastes compute on non-informative samples. |
> > | **DAPO** | 1004s | 775s | High cost due to re-generation. |
> > | **CLPO** | 698s | **631s** | **Becomes faster over time.** As the model improves, fewer problems need restructuring. |
> >
> > ### 5. Discussion of Related Work: Auto-Curriculum (Weakness 5)
> >
> > **Concern:** The discussion needs better contextualization regarding self-play and auto-curriculum.
> >
> > **Response:**
> > We will expand the "Related Work" section to explicitly contrast CLPO with existing Auto-Curriculum methods along two key dimensions:
> >
> > 1.  **Human Expectation vs. Exploration Adaptation:**
> >     *   *Traditional Auto-Curriculum:* Often relies on **human expectations** of a learning curve, pre-sorting data (e.g., "learn short sequences, then long ones"). This assumes a predictable trajectory.
> >     *   *CLPO:* RL training is inherently **stochastic and exploratory**. The model's actual progress often defies pre-set schedules. CLPO defines difficulty strictly via **real-time on-policy performance**. This "model-centric" approach is more robust to the unpredictable nature of RL exploration than "human-centric" pre-sorting.
> >
> > 2.  **Re-weighting vs. Active Restructuring:**
> >     *   *Traditional Methods:* Typically only change **sampling weights** (re-weighting). The dataset itself remains static.
> >     *   *CLPO:* Introduces **Active Restructuring**. Instead of just selecting problems, CLPO acts as a teacher that **rewrites** them—simplifying hard tasks that are currently out of reach, and diversifying medium tasks to prevent overfitting. This shift from "passive selection" to "active creation" is a fundamental novelty of our work.
> >
> > ---
> >
> > We hope these additional experiments (especially on ACPBench and MMLU-Pro) and clarifications address your concerns. We are confident that CLPO offers a robust, efficient, and scalable contribution to the field.

---

### Author Response · Authors · 2025-12-03
**Authors final summary**

---

## Summary of Strengths

All reviewers recognize the innovation and effectiveness of CLPO, with consensus on its academic value and practical performance in enhancing LLM reasoning via RLVR:

| Strength | Reviewers | Key Quotes & Observations |
|----------|-----------|---------------------------|
| **Novel Framework Design** | s6og, syXH | "Guided Self-Evolution" integrating curriculum learning with RLVR dynamically is a "novel integration" that "directly addresses uniform sampling limitations in RLVR" |
| **Effective Technical Mechanisms** | s6og, rCGW | Adaptive problem restructuring, difficulty-aware KL regularization, and online curriculum. rCGW confirms the RL paradigm "runs automatically" |
| **Strong Empirical Results** | h1ux, rCGW | SOTA performance across eight reasoning benchmarks with consistent improvements over GRPO and DAPO; "thorough evaluation proving effectiveness" |
| **Practical Value** | h1ux, syXH | Code "enhances reproducibility and facilitates further research"; "strong empirical validation through extensive experiments" makes the work impactful |

---

## Our Resolution of Reviewers' Major Concerns

We have addressed concerns about **dataset generalization**, **model scalability**, **computational efficiency**, and **novelty** through supplementary experiments, mechanism clarification, and code optimization. All revisions will be integrated into the manuscript.

---

### 1. Dataset Selection & Generalization
**Raised by:** s6og, rCGW

**Core Concerns:** Need for further validation on AIME25/GPQA Main/ACPBench; uneven improvements on MATH-500 and AMC23.

#### Resolutions with Evidence:

**Supplementary Experiments on Key Benchmarks:**

| Benchmark | Result | Significance |
|-----------|--------|--------------|
| **AIME25** | Already included in original study (Figure 5) | Performance consistent with AIME2024, showing stable improvements |
| **GPQA Main** | CLPO: **0.5737** vs. DAPO: 0.5022 | Significant gains beyond the standard Diamond subset |
| **ACPBench** | CLPO surpasses DAPO on all 14 sub-tasks | Representative gains: **+27.7%** on acp_app_bool, **+32.3%** on acp_just_bool |
| **HeadQA Equivalent** | MMLU-Pro (Table 1): CLPO achieves **+2.3%** over DAPO | Proves generalization beyond mathematical reasoning |

**Explanation of Uneven Improvements:**

- **MATH-500:** "Saturation effect"—baselines already reach 89%-91% accuracy (near Qwen3-8B's capacity ceiling). CLPO achieves **89.60%** (on par with SOTA) without degradation, while securing large gains on harder benchmarks like AIME.
- **AMC23:** Small dataset (~40 problems) dominated by "Easy" tasks. CLPO down-weights mastered samples to focus on high-value tasks, resulting in naturally smaller improvement margins.
- **MMLU-Pro:** As a knowledge-intensive benchmark (vs. pure reasoning), RLVR methods generally show limited gains, but CLPO still achieves stable positive improvements, demonstrating robustness.

---

### 2. Model Scalability & Adaptability
**Raised by:** s6og, h1ux, rCGW, syXH

**Core Concerns:** Initial experiments limited to Qwen3-8B-Base; requests for validation across model architectures/sizes; questions about Qwen3-8B-Base's restructuring capability and Qwen2.5-7B-Instruct performance.

#### Resolutions with Evidence:

**Cross-Architecture/Scale Validation (AIME24 Results):**

| Model | Configuration | CLPO | GRPO | DAPO |
|-------|--------------|------|------|------|
| **Llama-3.1-8B-Instruct** | 100 steps, acc/mean@32 | **0.039** | 0.005 | 0.017 |
| **Qwen3-32B** | 50 steps | **0.447** | 0.428 | — |

**Qwen Series Adaptability:**

- **Qwen3-8B-Base Capability:** Despite lacking chat instruction tuning, its strong reasoning performance (89.84% on GSM8K, 60.80% on MATH per Qwen3 Technical Report) enables valid problem restructuring, verified by Figure 1's example outputs.
- **Qwen2.5-7B-Instruct:** New experiments (120 steps on 8×H20 GPUs) show CLPO's advantages:
  - AIME2025: **0.089** vs. DAPO 0.072
  - AMC23: **0.566** vs. DAPO 0.521

---

### 3. Computational Efficiency & Overhead
**Raised by:** s6og, syXH

**Core Concerns:** Computational overhead from restructuring; efficiency comparison with baselines under equal budget.

#### Resolutions with Evidence:

CLPO realizes **"computational structural reallocation"** instead of adding redundant overhead. Controlled experiments on 8×H20 GPUs (Batch Size=64):

| Method | Step 1 Time (s) | Step 100 Time (s) | Key Analysis |
|--------|-----------------|-------------------|--------------|
| GRPO | 641 | 835 | Constant cost; wastes compute on mastered/hopeless samples |
| DAPO | 1004 | 775 | High initial cost due to frequent re-generation of discarded samples |
| **CLPO** | 698 | **631** | Faster over time; decreasing restructuring needs as model improves |

> Under fixed time/FLOPs budget, CLPO completes more effective update steps by focusing compute on high-value "medium" tasks, confirming its practical efficiency.

---

---

> ### Author Response · Authors · 2025-12-03
> **Authors final summary**
>
> ### 4. Technical Mechanism Clarification
> **Raised by:** s6og, syXH
>
> #### 4.1 Dynamic KL Regularization Stability
>
> **Concern:** Dynamic KL results seem noisy/unstable.
>
> **Resolution:** Figure 4 reflects **robustness** rather than instability:
> 1. Training curves for different α values are smooth and monotonic → stable optimization
> 2. All α converge to similar high performance on AIME2024 → hyperparameter insensitivity
> 3. Minor fluctuations stem from AIME's small sample size (30 problems), not algorithm defects
> 4. A fixed α is used across datasets/models without tuning → ease of use verified
>
> #### 4.2 Reward Design & Answer Preservation
>
> **Concerns:** How to guarantee answer consistency in restructuring; RL reward mechanism design.
>
> **Resolutions:**
>
> - **Answer Preservation:** Three-layer protection:
>   1. Prompt constraints
>   2. Original labels used
>   3. No contradictions in spot checks
>
> - **Reward Mechanism:** Strict outcome-based design:
>   ```
>   R(y, y*) = 1  if generated response y matches ground truth y*
>   R(y, y*) = -1 otherwise
>   ```
>
> ---
>
> ### 5. Novelty & Related Work Positioning
> **Raised by:** h1ux, rCGW, syXH
>
> **Core Concern:** Suspicions of incremental innovation; requests for clear distinction from prior work.
>
> #### Resolution:
>
> CLPO represents a **paradigm shift** in RLVR training with three fundamental innovations:
>
> | Dimension | Prior Work | CLPO Innovation |
> |-----------|-----------|-----------------|
> | **Curriculum Form** | Static offline pre-sorting (Auto-Curriculum, WizardLM) | **Policy-driven online curriculum**, dynamically adjusting via real-time on-policy rewards |
> | **Data Processing** | Passive sampling/re-weighting (DAPO, Self-Instruct) | **Active problem restructuring** (diversifying medium tasks, simplifying hard tasks) |
> | **Optimization Strategy** | Uniform constraints (DAPO's global KL regularization) | **Difficulty-aware optimization** with adaptive KL constraints for different task levels |
>
> ---
>
> ### 6. Code Usability & Reproducibility
> **Raised by:** h1ux
>
> **Concern:** Poor code readability and lack of documentation hinder reproduction.
>
> #### Resolution Plan (Camera-Ready Version):
>
> The codebase will be fully upgraded with:
>
> - **Comprehensive README.md** covering environment setup (Conda/Docker), data preparation, and training loop instructions
> - **One-click reproduction scripts** for core results (e.g., Qwen3-8B-Base experiments in Table 1)
> - **Refactored core modules** (Online Curriculum Builder, Restructuring, Dynamic KL) with detailed comments

---

### Note · Authors · 2026-01-26

I have read and agree with the venue's withdrawal policy on behalf of myself and my co-authors.

---

### Meta-Review · Area_Chair_dfrH · 2025-12-23

**Summary:**

This paper proposes a reinforcement learning framework named CLPO to enhance training efficiency for reasoning tasks. Three reviewers submitted reviews and participated in discussions, with initial scores leaning toward rejection. One reviewer decided to maintain their score after discussions. After thoroughly reviewing the paper and the authors' responses, the AC found that some reviewer concerns remained unresolved, such as performance across different models, reward mechanism design, and limited innovation. The AC recommends that the authors revise and submit to a subsequent conference.

**Reviewer Concerns:**

I believe some of the specific issues concerning the experiment have been resolved, while others remain unresolved.

**Reviewer Scores:**

They will maintain their scores.

---

### Decision · Program_Chairs · 2026-01-26

Reject